# Automorphic Equivalence-aware Graph Neural Network

**Fengli Xu**[1], **Quanming Yao**[1,2], **Pan Hui**[3], **Yong Li**[1]

[1]BNRIST & EE Department, Tsinghua University
[2]4Paradigm Inc.
[3]CSE Department, HKUST.
liyong07@tsinghua.edu.cn

## Abstract

Distinguishing the automorphic equivalence of nodes in a graph plays an essential role in many scientific domains, *e.g.*, computational biologist and social network analysis. However, existing graph neural networks (GNNs) fail to capture such an important property. To make GNN aware of automorphic equivalence, we first introduce a localized variant of this concept — ego-centered automorphic equivalence (Ego-AE). Then, we design a novel variant of GNN, *i.e.*, GRAPE, that uses learnable AE-aware aggregators to explicitly differentiate the Ego-AE of each node's neighbors with the aids of various subgraph templates. While the design of subgraph templates can be hard, we further propose a genetic algorithm to automatically search them from graph data. Moreover, we theoretically prove that GRAPE is expressive in terms of generating distinct representations for nodes with different Ego-AE features, which fills in a fundamental gap of existing GNN variants. Finally, we empirically validate our model on eight real-world graph data, including social network, e-commerce co-purchase network, and citation network, and show that it consistently outperforms existing GNNs. The source code is public available at `https://github.com/tsinghua-fib-lab/GRAPE`.

## 1 Introduction

The past few years have witnessed the phenomenal success of GNNs in numerous graph learning tasks, such as node classification [24], link prediction [59], and community detection [19], which is largely due to their capability of simultaneously modelling the connecting patterns and feature distribution in each node's local neighborhood. As a result, it leads to a surge of interests from both academia and industry to develop more powerful GNN models [52]. Despite of various architectures, the most popular GNNs, like GCN [24], GraphSAGE [19], GAT [48] and GIN [55], apply permutation invariant aggregate function on each node's local neighborhood to learn node embeddings, which leads to concerns about their representational power [17, 55].

In this paper, we investigate GNN's expressiveness from an important but largely overlooked angle, *i.e.*, the capacity to distinguish automorphic equivalence within each node's local neighborhood. Automorphic equivalence (AE) [14] is a classic concept to differentiate the structural role of each node in a given graph. Specifically, two nodes are considered to be AE only if they are interchangeable in some index permutations that preserve the connection matrix, *i.e.*, graph automorphisms [35]. AE can identify the nodes that exhibit identical structural features in a graph, which makes it a central topic in computational biologist, social network analysis and other scientific domains [33, 34]. For example, empirical studies show AE is an important indicator of social position and behavior similarity in social network [16, 37], which thus might significantly benefit GNN architecture design.

35th Conference on Neural Information Processing Systems (NeurIPS 2021).

Empirically efficient heuristics have been proposed to identify the AE in moderate scale graphs by enumerating all the possible automorphisms [6, 45]. However, previous analytic methods only classify nodes into categorical equivalence sets. Although categorical features can be jointly optimized with GNNs under various frameworks [30, 54], little previous efforts are invested to principally incorporate AE into GNNs. Thus, we aim to design a novel GNN model that is provably expressive in capturing AE features and can be tuned in a data-dependent manner based on the graph data and targeted applications, which will effectively allow us to learn expressive function to harness the power of AE feature.

Here, we propose GRaph AutomorPhic Equivalent network, *i.e.*, GRAPE, a novel variant of GNN that can learn expressive representation by differentiating the automorphic equivalences of each node's neighbors. First, GRAPE extends the classic AE concept into a localized setting, *i.e.*, Ego-AE, to accommodate the local nature of GNNs. Specifically, Ego-AE identifies the local neighborhoods of each node by mapping with given subgraph templates and then partitions the neighboring nodes into Ego-AE sets based on the graph automorphisms in neighborhood. Second, we design a learnable AE-aware aggregators to model the node features in these Ego-AE sets, which adaptively assigns different weights to neighboring nodes in different Ego-AE sets and explicitly model the interdependency among them. Moreover, in order to capture complex structural features, GRAPE proposes to fuse the embeddings learned from Ego-AE sets identified by different subgraph templates with a squeeze-and-excitation module [22]. Finally, to alleviate the barrier of subgraph template design, we propose an efficient genetic algorithm to automatically search for optimal subgraph templates. Specifically, it gradually optimizes a randomly initiated population of subgraph templates by iteratively exploring the adjacency of good performing candidates and eliminating the bad performing ones. To accelerate the search process, we further design an incremental subgraph matching algorithm that can leverage the similarity between subgraphs to greatly reduce the complexity of finding matched instances.

We theoretically prove that the proposed GRAPE is expressive in terms of learning distinct representations for nodes with different Ego-AE sets, which fundamentally makes up the shortcomings of popular GNN variants, *e.g.*, GCN [24], GraphSAGE [19], GAT [48] and GIN [55]. Moreover, we empirically validate GRAPE on eight real-world datasets, which cover the scenarios of social network, citation network and e-commerce co-purchase network. Experiments show GRAPE is consistently the best performing model across all datasets with up to 26.7% accuracy improvement. Besides, case studies indicate GRAPE can effectively differentiate the structural roles of each node's neighbors. Moreover, the proposed genetic algorithm efficiently generates high quality subgraph templates that have comparable performance with the hand-crafted ones.

## 2    Related Works

In the sequel, we define a graph as $G = (\mathcal{V}, \mathcal{E})$, where $\mathcal{V} = \{v_1, \cdots, v_n\}$ is the set of nodes and $\mathcal{E} = \{(v_i, v_j)\}$ is the set of edges. Let the feature vector of node $v_i$ be $\mathcal{X}(v_i)$, and $\mathcal{N}(v_i)$ represents the set of $v_i$'s neighbor nodes. $N$ is the number of node and $M$ is the embedding size.

### 2.1    Automorphic Equivalence (AE) in Graph Analysis

Here, we investigate one of the most popular structural equivalence concepts, *i.e.,* automorphic equivalence (AE) [14] which plays a central role in computational biologist, social network analysis and other scientific domains [33, 34]. The most interesting part of AE is that it identifies the nodes with exact same structural patterns, *e.g.*, degree and centrality [15], but not necessarily connecting to the same neighboring nodes. Basically, AE is defined as follows.

**Definition 2.1.** *Given a graph $G = (\mathcal{V}, \mathcal{E})$, an **automorphism** $\pi(\star)$ is a node permutation that preserves the adjacency matrix, i.e., the permuted nodes $\pi(v_a)$ and $\pi(v_b)$ are connected if and only if nodes $v_a$ and $v_b$ are connected. Two nodes $v_a$ and $v_b$ are considered to be **__Automorphic Equivalence__ (AE)** if there is a graph automorphism that maps one onto the other, i.e., $\pi(v_a) = v_b$.*

Nodes in $\mathcal{V}$ that are AE with each other constitutes a AE set. An example of such sets is in Figure 1 (a). Specifically, AE sets can be identified by enumerating the automorphism group of the given graph with efficient *Nauty* algorithm [35], and the nodes that are mapped onto each others in different automorphisms will be partitioned into same AE sets.

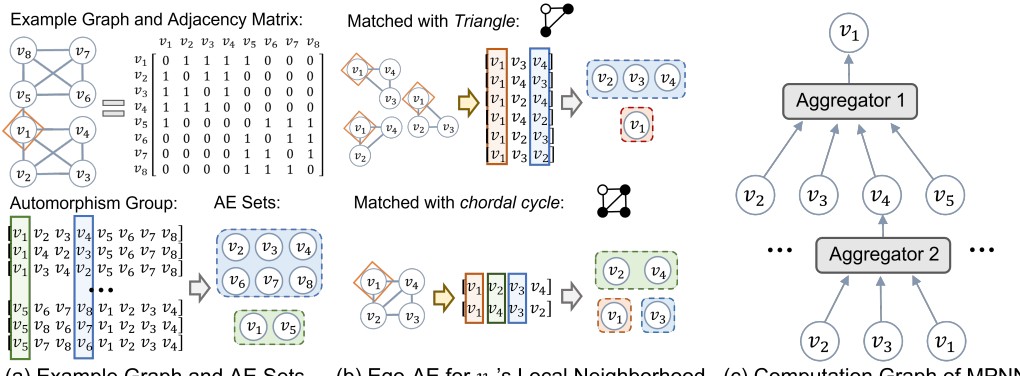

(a) Example Graph and AE Sets    (b) Ego-AE for $v_1$'s Local Neighborhood   (c) Computation Graph of MPNN

Figure 1: Illustration of AE and the limitation of MPNN framework.

Previous works have attempted to preserve the structural similarities on graph via various node embedding algorithms [12, 2, 40]. Recently, GraphWave [12] was proposed to leverage wavelet diffusion patterns to capture the structural roles in node representations. Besides, role2vec [2] introduced a generalized feature-based random walks that aims to represent the structural similarities among nodes.

However, the connection between AE and GNN has not been examined in existing literature. Moreover, AE is defined on whole graph level, which is infeasible to compute in large graphs and goes against the inherent local nature of most GNN frameworks. In this paper, we aim to extend the AE concept to local setting, and propose a novel GNN model to harness its power.

## 2.2 The Expressive Power of GNNs

The recent success of GNNs draws increasing interests in investigating their capability in capturing structural properties [55, 17]. Specifically, message passing neural network (MPNN) is the most popular GNN framework [18, 19, 55].

Let AGG be the *aggregate* function that collects feature from neighbors, and COMB be the *combine* function that integrates each node's self feature with those from AGG. Generally, MPNN generates representation for a node $v_t$ at $k$-th layer as

$$\boldsymbol{h}_{v_t}^k = \mathsf{COMB}\big(\boldsymbol{h}_{v_t}^{k-1},\ \mathsf{AGG}(\mathcal{H}_{v_t}^{k-1})\big), \tag{1}$$

where $\boldsymbol{h}^0(v_i) = \mathcal{X}(v_i)$, and $\mathcal{H}_{v_t}^{k-1} = \big\{\boldsymbol{h}_{v_j}^{k-1}|v_j \in \mathcal{N}(v_t)\big\}$. Existing MPNNs use local permutation invariant AGGs to compute node embeddings, which subsumes a large class of popular GNN models such as GCN [24], GraphSAGE [19], GAT [48], Geniepath [31] and GIN [55]. The family of MPNNs is proven to be theoretically linked to Weisfeiler-Lehman (WL) subtree kernel [19]. Subsequently, they are at most as powerful as 1-WL test on discriminating graph isomorphisms [55].

More recently, several attempts have been made to improve the expressiveness of GNN beyond MPNN framework. They can mainly be classified into two categories: augmenting node features and designing more powerful architectures. In terms of augmenting node feature, recent works proposed to introduce various additional feature [42, 25, 59]. 3D-GCN uses additional 3D point cloud feature to differentiate neighbors and facilitate learnable graph kernels [29]. Moreover, GNN variants can in theory achieve universal approximation on graph by equipping nodes with randomly initialized feature vector [43], but they are difficult to generalize to different graphs in practice [5]. Previous work also proposed to augment node feature with substructure count [5], which however cannot reveal the local structural roles in each node's neighborhood. On the other hand, the previous efforts of designing more powerful GNN architecture are dedicated to different structural properties, *e.g.*, graph isomorphism [55] and graph moment [11]. Although AE is an important concept for graph data analysis, it has not been addressed by previous GNN research. More detailed comparison with the existing GNN variants is provided in Appendix A.

However, previous works have shown that identifying automorphic equivalence is a strictly more difficult task than discriminating graph isomorphisms [46]. Specifically, two AE nodes always have

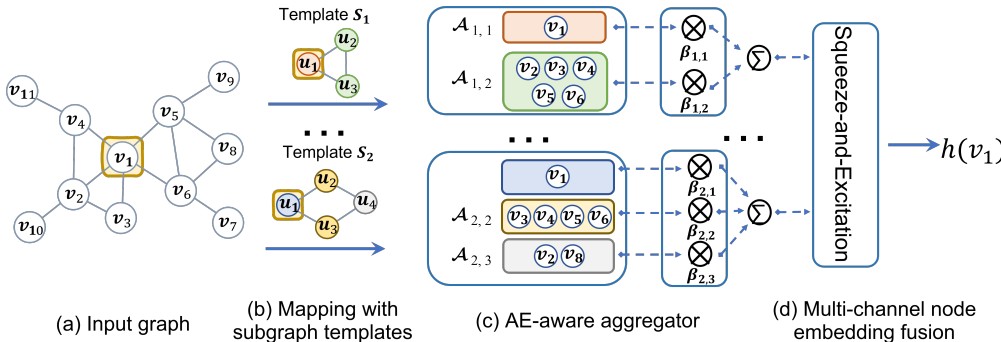

Template $S_1$    Template $S_2$

(a) Input graph    (b) Mapping with subgraph templates    (c) AE-aware aggregator    (d) Multi-channel node embedding fusion

Figure 2: The GRAPE model. (a) The input graph with node $v_1$ as ego node. (b) Mapping node $v_1$'s neighborhood with given subgraph templates, where nodes with same color are Ego-AE. (c) Aggregating features from AE sets with learnable AE-aware aggregators. (d) Fusing multi-channel node embedding with squeeze-and-excitation module.

isomorphic neighborhood, while the nodes with isomorphic neighborhood are not necessary AE [15]. Therefore, it raises concerns about MPNN's expressive power of AE feature, which has not been adequately investigated in previous research. In this paper, we aim to design a novel GNN model that is provably expressive in modeling Ego-AE, which falling in the category of designing novel GNN architecture. To the best of our knowledge, we are the first to empower GNN with the capability of capturing automorphic equivalence.

## 2.3 Genetic Algorithm

Genetic algorithm [13] is a widely adopted algorithm for combinatorial optimization problems, *e.g.*, traveling salesman problem. Recently, it has also been used to tune hyper-parameters [32] and search neural architectures for deep networks [57, 53]. Basically, genetic algorithm mimics the natural selection process to iteratively to search for better solutions by exploring the adjacency of promising candidates and eliminating the worst-performing ones [10]. Therefore, it can iteratively optimize the candidate population from *parent generation* to *children generation*. Specifically, genetic algorithms are often made up the following four components: (i) *Mutation*: explores the adjacency of promising candidates in *parent generation* by generating slightly different candidates in *children generation*; (ii) *Crossover*: search for different combinations of genetic features in the candidates in *parent generation*; (iii) *Evaluation*: measure the fitness of candidates in given tasks; (iv) *Selection*: eliminating the candidates with worst performance.

Here, we design a genetic algorithm to automatically optimize the subgraph templates in the proposed GRAPE. It effectively allows us to search the architecture of GRAPE in a data-dependent manner, which significantly reduces the barrier of subgraph template design.

## 3 The Proposed Method

One prominent feature of most GNN variants is that the node embedding are generated based on the local neighborhood, which significantly improved the scalability and generalization of GNN models. To accommodate the local nature of GNNs, the concept of AE needs to be fundamentally extended and redefined on each node's local neighborhood. Besides, the local neighborhoods defined by different subgraph templates may exert different influence on the ego node [58, 2]. Thus, we propose a subgraph template-dependent local version of AE, *i.e.*, ego-centered automorphic equivalence (Ego-AE). Specifically, a subgraph template is defined as a connected graphlet $S = (\mathcal{U}, \mathcal{R})$, where $\mathcal{U}$ and $\mathcal{R}$ are the sets of nodes and edges, respectively. To differentiate the unique role of ego node, we set an anchoring node in subgraph template that always maps to the ego node. Given a graph $G = (\mathcal{V}, \mathcal{E})$, a subgraph template $S = (\mathcal{U}, \mathcal{R})$ and a node $v_t$, the Ego-AE on $v_t$'s local neighborhood is defined as follow.

**Definition 3.1.** *We define $\mathcal{M}_S(v_t)$ as the set of subgraphs that match the subgraph template $S$ in $v_t$'s local neighborhood. An ego-centered automorphism $\pi_e(\star)$ is an automorphism on the matched*

subgraphs $m \in \mathcal{M}_S(v_t)$ that has a fixed index of node $v_t$, i.e., $\pi_e(v_t) \equiv v_t$. Two nodes $v_a$ and $v_b$ are considered to be **Ego-centered Automorphic Equivalence (Ego-AE)** if there exists an automorphism $\pi_e(\star)$ that maps one onto the other, i.e., $\pi_e(v_a) = v_b$.

Without loss of generality, various forms of subgraph template $S$ can be adopted to capture the structural patterns of different semantics. Figure 1 (b) shows the Ego-AE on $v_1$'s local neighborhoods with the subgraph templates of *triangle* and *chordal cycle*. Specifically, we first identify all the matched subgraphs with the anchoring nodes (white color) in subgraph templates fixed to the ego node $v_1$, and then the nodes covered by the matched instances are partitioned into Ego-AE sets based on the corresponding automorphisms. We can observe that Ego-AE successfully differentiates the roles of $v_1$'s neighboring nodes based on the structural features.

However, GNN's capacity in capturing Ego-AE is largely unknown in the literature. In fact, we prove that the standard MPNN has fundamental limitations (see Section 3.3), while other GNN variants focus on different graph properties which largely overlooked Ego-AE.

### 3.1 The Proposed GRAPE Model

We aim to propose a novel GNN model, *i.e.*, GRaph AutomorPhic Equivalence network (GRAPE), that is provably expressive in capturing Ego-AE. The overall framework is in Figure 2. In the sequel, we describe them in details, and the complete algorithm of GRAPE is in Appendix C.1.

#### 3.1.1 AE-aware Aggregator

Here, we design a novel AE-aware aggregators to learn from the Ego-AE sets in each node's local neighborhood with given subgraph templates. Specifically, we denote $v_t$'s Ego-AE sets with subgraph template $S_l$ as $\mathcal{T}_l = \{\mathcal{A}_{l,1}(v_t), ..., \mathcal{A}_{l,j}(v_t), ..., \mathcal{A}_{l,m_l}(v_t)\}$, where $\mathcal{A}_{l,j}(v_t)$ is the set of nodes corresponding to the $j$-th sets of Ego-AE nodes in $S_l$ and $m_l$ is the total number Ego-AE sets. Then, $v_t$'s node embedding $\boldsymbol{h}_l^k(v_t)$ can be computed as follows:

$$\boldsymbol{h}_l^k(v) = \mathsf{MLP}\big(\sum\nolimits_j \beta_{l,j} \cdot \sum\nolimits_{v_n \in \mathcal{A}_{l,j}(v_t)} \boldsymbol{h}_l^{k-1}(v_n)\big), \tag{2}$$

where $\beta_{l,j}$'s are learnable weights that model the importance of $\mathcal{A}_{l,j}(v_t)$ and $\mathsf{MLP}(\cdot)$ is a multi-layer perception (MLP) function [21] that generates output embeddings.

Equation (2) is illustrated in Figure 2 (c). Differs from MPNNs, the proposed AE-aware aggregator can explicitly differentiate the neighboring nodes with different structural roles by assigning different weights $\beta_{l,j}$ to them. It allows GRAPE to capture the combination of important Ego-AE sets and effectively models the interdenpendency among them. Note that GRAPE does not have an explicit *COMBINE* function to account for the ego node's self feature, since the ego node $v_t$ will always be captured in a unique Ego-AE set, such as the $\mathcal{A}_{1,1}$ and $\mathcal{A}_{2,1}$ in Figure 2 (c).

#### 3.1.2 Fusing Embeddings from Different Aggregators

To simultaneously capture different structural feature, we design a squeeze-and-excitation module to fuse the node embeddings learned from a set of subgraph templates, which is inspired by the channel-wise enhancement technique recently proposed in [22]. Specifically, by leveraging a set of subgraph templates $\Omega = \{S_1, S_2, ..., S_L\}$, GRAPE can learn multiple AE-aware aggregators with each subgraph template to capture different structural features respectively. The GRAPE can learn to assign different weights $\boldsymbol{\alpha}^k$ for $\boldsymbol{h}_l^k(v_t)$ and generate the fused embedding for $v_t$ as following

$$\boldsymbol{h}^k(v) = \sum\nolimits_{l \in 1, ..., L} \boldsymbol{\alpha}^k[l] \cdot \boldsymbol{h}_l^k(v). \tag{3}$$

Here, the learnable weights $\boldsymbol{\alpha}^k$ is computed as following

$$\boldsymbol{\gamma}^k[l] = \frac{1}{N} \sum\nolimits_{n=1}^N \mathsf{MEAN}(\boldsymbol{h}_l^k(v_n)), \quad \boldsymbol{\alpha}^k = \mathsf{ReLU}\big(\boldsymbol{W}_2^k \cdot \mathsf{ReLU}(\boldsymbol{W}_1^k \cdot \boldsymbol{\gamma}^k)\big), \tag{4}$$

where $\boldsymbol{\gamma}^k[l]$ is the global average pooling on the node embeddings learned with subgraph template $S_l$, $\boldsymbol{W}_1^k$ and $\boldsymbol{W}_2^k$ are two learnable matrices with $\mathbb{R}^{L \times L}$ size, and $\mathsf{ReLU}$ is the relu activation function.

---
**Algorithm 1** Genetic Algorithm for Subgraph Template Optimization
---
1: Input graph $G = (\mathcal{V}, \mathcal{E})$; node feature $\mathcal{X}(v)$; probability of edge mutation, node mutation, crossover = $p_e, p_n, p_c$; size of gene pool $B$; subgraphs per gene $L$; elimination size $Z$;
2: $genePool$ = InitPool($B$, $L$);
3: **for** $k \in 1, ..., K_2$ **do**
4:     $genePool$ = Mutate($genePool, p_e, p_n$);          /*Mutate Subgraph Templates*/
5:     $genePool$ = Crossover($genePool, p_c$);           /*Generate Different Combinations*/
6:     **for** $gene \in genePool$ **do**
7:        $\{\mathcal{T}_1, ..., \mathcal{T}_L\}$ = Match($G$, $gene$)         /*Match on Graph.*/
8:        $accuracy$ = $F(\{\mathcal{T}_1, ..., \mathcal{T}_L\}, \Theta)$        /*Evaluate Performance.*/
9:        $metricPool.append(accuracy)$
10:    **end for**
11:    $genePool$ = Select($genePool$, $metricPool$, $Z$)     /*Eliminate and Reproduce*/
12: **end for**
13: **return** Best performing $gene \in genePool$.
---

## 3.2 Genetic Search of Subgraph Templates

To reduce the barrier of hand-crafted subgraph templates, we formulate the automatic subgraph template design problem as an optimization problem that aims to search for the best performing combinations of subgraph templates $\Omega$. Let the designed GRAPE in Section 3.1 be $F$ with model parameter $\Theta$, which leverages the Ego-AE sets $\{\mathcal{T}_1, ..., \mathcal{T}_L\}$ identified by *Matching* $\Omega$ on the given graph $G$. This subsequently leads to the following bi-level optimization [7] problem:

$$\max_{\{\Omega|G\}} F\left(\{\mathcal{T}_1, ..., \mathcal{T}_L\}, \Theta^\star\right), \quad \text{s.t.} \quad \begin{cases} \{\mathcal{T}_1, ..., \mathcal{T}_L\} = \text{Match}(G, \Omega) \\ \Theta^\star = \arg\max_\Theta F(\{\mathcal{T}_1, ..., \mathcal{T}_L\}, \Theta) \end{cases}, \quad (5)$$

However, the proposed optimization problem is difficult mainly for two reasons: 1) the search space of subgraph templates is discrete and not differentiable; 2) matching subgraph templates in a large graph is computationally expensive.

Our key intuition to address these challenges is that similar subgraph templates often have slightly different pools of matched instances, which is likely to result in similar model performance. Therefore, by gradually exploring the adjacent space of good performing subgraph templates we can effectively avoid bad candidates. This inspires us to design a genetic optimization framework, which can navigate through the discrete search space via the gradual mutations between generations. Moreover, the similarities between iteratively searched subgraphs can be further leveraged to design efficient subgraph matching algorithm. The details are described as follows.

### 3.2.1 Genetic Subgraph Template Search

We define gene population as a set of $B$ *genes*, where each gene is a set of $L$ subgraph templates. The gene population is initiated as the most basic subgraph templates, *i.e.*, edge. Then, the gene population is optimized through $K_2$ rounds of genetic operations, which consists of *mutate*, *crossover*, *evaluate* and *select*. The *mutate* operation allows us to explore slightly more complex subgraph templates, *i.e.*, the subgraph templates with one randomly added nodes or edges, which are denoted as *children subgraphs*. Besides, the *crossover* operation will randomly exchange some subgraph templates between two genes, which allows us to try different combinations of subgraph templates. Moreover, the *evaluate* and *select* operations will identify and remove the worst-performing $Z$ genes and reproduce the best-performing genes. Finally, we use the subgraph templates encoded in the best performing *gene* as the input of $F$. Thus, these operations allow us to gradually explore the adjacent search space of promising *genes* and automatically optimize the subgraph templates. The genetic algorithm is presented in Algorithm 1.

### 3.2.2 Efficient Subgraph Template Matching

We have the following Proposition 3.1 for the matched instances of the mutated *children subgraph*, which can be leveraged to accelerate the $Match$ function in Algorithm 1. The proof is straightforward since the *children subgraphs* are extended from *parent subgraphs* by randomly adding one node or

edge. In fact, the adding edge *mutation* effectively acts as a filtering mechanism on matched instances, and the adding node *mutation* effectively grows the matched instances of *parent subgraph*. Therefore, instead of computing the matched instances of *children subgraph* from scratch, the *mutate* operation facilitates us to save significantly amount of computation by reusing and extending the matched instances of the corresponding *parent subgraphs*. As a result, we propose an incremental subgraph matching algorithm to leverage this proposition to accelerate subgraph matching process, which is illustrated in Appendix C.2 in details.

**Proposition 3.1.** *Given a graph $G$, a parent subgraph $S_p$ and a children subgraph $S_c$, we denote $S_p$ and $S_c$'s match instances set as $\mathcal{M}_p$ and $\mathcal{M}_c$, respectively. Then, we have $m_p \subset m_c$: $\exists m_p \in \mathcal{M}_p$, for $\forall m_c \in \mathcal{M}_c$. That is the matched instances of children subgraphs $m_c$ will always contain a matched instance of parent subgraphs $m_p$. Thus, $m_c$ can be efficiently identified by incrementally extending $m_p$.*

## 3.3 Theoretical Analysis

Here, we aim to answer two questions: 1) how does the expressiveness of AE-aware aggregator relate to previous works; and 2) is our designed AE-aware aggregator expressive enough to capture Ego-AE feature. Previous researches mainly investigate the expressiveness of GNN through the scope of graph isomorphism test, while it is expressive power on capturing AE feature is largely unknown. Specifically, we have the following proposition about the limitations of standard MPNNs.

**Proposition 3.2.** *There exist graphs that have different Ego-AE sets for a given node, but MPNNs in (1) with arbitrary number of layers and hidden units cannot distinguish them.*

We provide a constructive proof in Appendix B.1. Therefore, this proposition shows MPNNs have fundamental limitations in modeling the structural role of each node's neighbors. On the contrary, we have the following theorem about the expressive power of the proposed AE-aware aggregator. The theoretical proof of this theorem is provided in Appendix B.2. It shows our AE-aware aggregator is provably expressive in capturing Ego-AE feature.

**Theorem 3.1.** *For countable feature space $\mathcal{X}$, let $v_a$ and $v_b$ be two nodes with different Ego-AE sets. The AE-aware aggregator in (2) can discriminate two nodes with learned distinct embeddings.*

**Remark 1.** *Previous works on GNN's structural feature expressiveness mainly follow the hierarchy of neighborhood isomorphic,* i.e.*, can the GNNs differentiate two nodes that have different isomorphisms in their local neighborhoods. However, graph automorphism is a special isomorphism that maps a graph on to itself [35]. Therefore, Ego-AE is a stricter structural condition than neighborhood isomorphic. That is two automorphically equivalent nodes are always neighborhood isomorphic, while the converse statement is false even if the local neighborhoods expand to the entire graph [15]. As a result, previous works that aim to differentiate neighborhood isomorphic nodes with various forms of WL tests,* e.g.*, GIN [55] and k-GNN [38], cannot capture Ego-AE feature. Our GRAPE model aims to fill in this gap.*

# 4 Experiments

**Datasets.** Three types of real-world datasets are used, *i.e.*, academic citation networks, social networks and e-commerce co-purchase network.

- *Citation networks* [44]: We consider 2 widely used citation networks, *i.e.* Cora and Citeseer. In these datasets, nodes represent academic papers and (undirected) edges denote the citation links between them. Following the setting in previous work [24], we use each paper's bag-of-words vector as its node feature and the subject as its label.

- *Social networks* [47]: We use 5 real-world social networks which are collected from the Facebook friendships within 5 universities, *i.e.*, Hamilton, Lehigh, Rochester, Johns Hopkins (JHU) and Amherst. The nodes represent students and faculties. Besides, we use one-hot encoding of their gender and major as node feature, and set the labels as their enrollment years.

- *E-commerce co-purchase networks* [26]: This dataset was collected by crawling the *music* items in Amazon website. If an item $a$ is frequently co-purchased with $b$, the graph contains a directed edge from $a$ to $b$. We use the average ratings of items as node labels, and we set the node features as the number of reviews and downloads.

**Compared Methods.** We compare our GRAPE with state-of-the-art GNN models, including GCN [24], GraphSAGE [19], GIN [55], GAT [48], Geniepath [31], Mixhop [1], Meta-GNN [41] and DE-GNN [27]. Specifically, GCN and GraphSAGE are two most popular GNN variants, and GIN is customized to better capture structural property. Besides, GAT and Geniepath use the attention mechanism to learn adaptive neighborhood for each node. Moreover, as more recent baselines, Mixhop, Meta-GNN and DE-GNN learn node embeddings from higher-order structural information. Specifically, Mixhop proposes difference operators on different hops of neighbors; Meta-GNN leverages predefined subgraphs to identify higher-order neighborhood; DE-GNN encodes the shortest path distance among nodes. To ensure fair comparison, we follow the optimal architectures as described in previous works, and we use the official implementations released by the authors or integrated in Pytorch platform [39].

Following the common design choices in previous works [24, 19, 48], we adopt a 2-layer architecture. The hyper-parameter tuning and detailed experiment settings are discussed in Appendix D.1. Based on the prior knowledge in related areas [3, 20], we design five subgraph templates ($S_\star$) for each domain of datasets respectively, which are described in Appendix D.2. We use same subgraph templates for Meta-GNN for fair comparison.

## 4.1 Benchmark Comparison

The classification accuracy of all methods are compared in Table 1. We observe that GRAPE is the best performing model across all datasets. Specifically, the performance gains are most prominent in social datasets. The improvements are smaller yet still significant on citation and E-commerce datasets. One plausible explanation is the structural features play more important roles in social network analysis. Following [49, 55], to investigate the influence of the node feature on the expressiveness of GNN, we also evaluate the models on datasets that use all-ones dummy node features and randomly initialized node features (see Appendix D.3 for details). We observe that GRAPE achieves consistent performance gains independent of node features, which echos the findings in previous studies that stronger topological feature can usually boost GNN's learning performance [55, 59, 49].

Table 1: Classification accuracy on datasets with original node feature (%). The best-performing GNNs are in boldface, and the second best ones are underlined.

| Model | Social | | | | | Citation | | Ecomm. |
| --- | --- | --- | --- | --- | --- | --- | --- | --- |
| | Hamilton | Lehigh | Rochester | JHU | Amherst | Cora | Citeseer | Amazon |
| GCN | 19.4±2.0 | 24.0±1.2 | 21.1±1.5 | 20.5±0.8 | 17.0±1.4 | **86.9±1.7** | **74.8±1.0** | 47.4±1.3 |
| GraphSAGE | 20.5±2.0 | 17.8±2.2 | 18.5±1.8 | 17.1±2.5 | 17.8±1.6 | 85.6±0.9 | 70.3±1.3 | 19.6±1.0 |
| GIN | 23.7±3.1 | 19.0±2.0 | 21.2±0.9 | 21.5±3.5 | 26.3±3.8 | 86.5±1.2 | 72.6±1.5 | 48.5±2.5 |
| GAT | 18.3±2.1 | 22.7±0.9 | 20.2±1.6 | 19.8±1.4 | 17.7±2.2 | **87.1±2.1** | **74.6±1.7** | 39.4±0.8 |
| Geniepath | 19.1±1.5 | 22.9±1.0 | 21.7±1.1 | 19.8±1.4 | 17.5±1.7 | 81.2±1.5 | 69.8±1.7 | 57.9±0.8 |
| Meta-GNN | 22.9±1.8 | 24.3±2.3 | 23.2±0.5 | 27.3±2.9 | 23.5±2.2 | 86.8±1.1 | 74.4±1.1 | 56.6±1.2 |
| Mixhop | 19.1±0.1 | 23.2±0.2 | 18.0±0.0 | 18.3±0.1 | 17.0±0.1 | 80.9±0.7 | 72.9±0.8 | 57.4±0.8 |
| DE-GNN | 21.7±2.5 | 24.9±2.1 | 18.0±0.0 | 19.8±1.7 | 18.9±2.2 | 31.9±0.0 | 39.1±2.3 | 58.1±0.1 |
| GRAPE | **28.1±2.1** | **27.3±3.8** | **25.0±1.8** | **34.6±1.3** | **32.6±2.2** | **87.1±1.8** | **74.6±1.5** | **58.6±0.4** |

We show the computation cost of GRAPE and exemplar GNNs in terms of wall-clock training time in Figure 3 (a). Note that both Geniepath and GAT leverage attention mechanism, which can only be trained on CPU (the training with GPU causes the out-of-memory error). Meta-GNN is coupled with complex graph sampling process and takes much more time to train. Thus, these methods are not plotted. We observe GRAPE takes comparable time to train on the example dataset as the classic GNN variants of Mixhop and GraphSAGE, which demonstrates the efficiency of our model. The theoretical time complexity analysis is provided in Appendix C.3.

Besides, we also analyze the learned squeeze-and-excitation weights on each subgraph template and Ego-AE set. From Figure 3 (b), we observe the weight is significantly skewed to edge template $S_1$ in first layer, while it distributes more evenly on more complex templates in the second layer. One plausible reason is that the model tend to keep the neighborhood large in the first hop to collect more

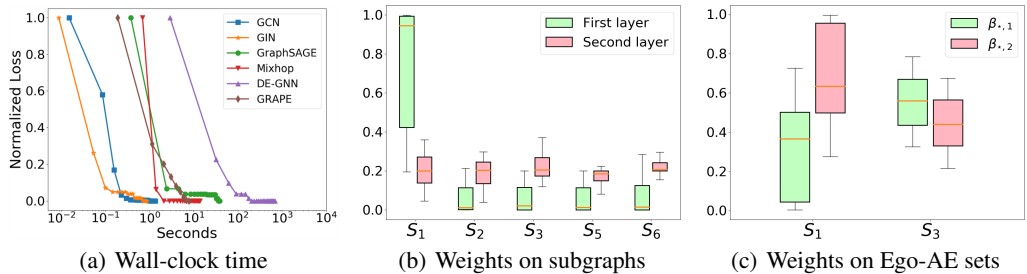

| (a) Wall-clock time | (b) Weights on subgraphs | (c) Weights on Ego-AE sets |

Figure 3: Illustrating the training time and the attention weights learned by GRAPE on *Lehigh* dataset.

feature. As a result, it assigns higher weights to the simplest template, *i.e.*, $S_1$ edge, in the first layer, while it fuses more diverse structural feature in the second layer by assigning more even weights to various subgraph templates. Figure 3 (c) shows the AE-aware aggregator can effectively distinguish the neighboring nodes with different structural roles. For example, in *Lehigh* dataset, GRAPE assigns lower weights to the Ego-AE set of ego node $\{u_1\}$ and higher weights to the AE set of neighbors $\{u_2\}$ on edge template $S_1$, while the weights distribute more evenly between $\{u_1\}$ and $\{u_2, u_3\}$ on triangle template $S_3$. It suggests the connected neighborhood nodes are more important than ego node on edge template $S_1$, while they have similar importance on triangle template $S_3$.

## 4.2 Case Study on AE-aware Aggregators

We conduct a case study to better understand how the AE-aware aggregators contribute to GRAPE. Specifically, we first randomly select a node $v_c$ in *Lehigh* dataset that is wrongly classified by all GNN variants except GRAPE, and then analyze its 2-hop neighborhood $\mathcal{N}^2(v_c)$ and the neighboring nodes that match triangle template $S_3$ and 4-clique template $S_5$ in Figure 4 and Table 2. We observe that there are 3,600 nodes in $\mathcal{N}^2(v_c)$ and they distribute evenly on multiple labels, which not only might confuse MPNNs but also tend to cause over-smoothness problem [28]. On the other hand, both templates $S_3$ and $S_5$ significantly reduce the neighborhood size, and the percentage of neighbors that have same labels as $v_c$ increases from 19.7% in $\mathcal{N}^2(v_c)$ (ranked $3^{rd}$) to 45.3%∼46.0% in $\mathcal{A}_{3,2}(v_c)$ and $\mathcal{A}_{5,2}(v_c)$ (ranked $1^{st}$). It shows the AE-aware aggregator can successfully capture the "social homophily" effect, where the nodes in tightly connected communities, *e.g.*, triangle and 4-clique structure, tend to have similar property [36].

Table 2: Number of nodes and most frequent labels in $v_c$'s 2-hop neighborhood and Ego-AE sets.

| | # nodes | The percentage of top 5 labels ($v_c$'s label is 11) |
|---|---|---|
| 2-hop neigh. $\mathcal{N}^2(v_c)$ | 3,600 | 13 (23.2%), 12 (21.8%), **11 (19.7%)**, 10 (17.3%), 14 (12.9%) |
| Ego-AE set $\mathcal{A}_{3,2}(v_c)$ | 46 | **11 (45.3%)**, 10 (17.2%), 12 (14.1%), 13 (12.5%), 9 (7.8%) |
| Ego-AE set $\mathcal{A}_{5,2}(v_c)$ | 222 | **11 (46.0%)**, 12 (24.9%), 10 (13.8%), 13 (11.0%), 9 (2.8%) |

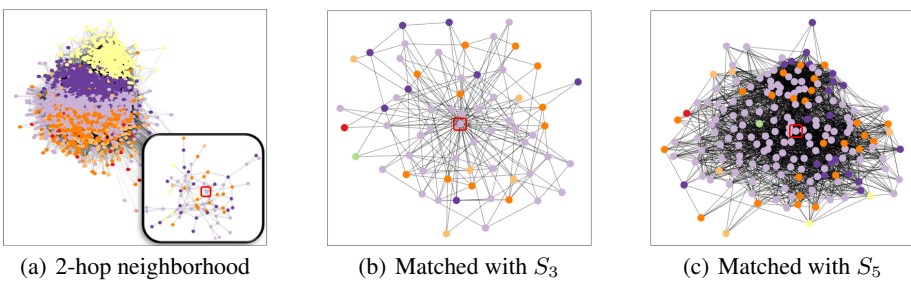

| (a) 2-hop neighborhood | (b) Matched with $S_3$ | (c) Matched with $S_5$ |

Figure 4: Case study of a node $v_c$ in *Lehigh* dataset. The ego node $v_c$ is positioned in the center and the inset of (a) shows 100 samples. Node colors represent the ground-truth label.

## 4.3 Subgraph Template Search

We evaluate the genetic subgraph template search algorithm on the social dataset. Specifically, we initialize GRAPE with three simplest subgraph template, *i.e.*, edges. We compare our algorithm with two baseline methods: 1) Genetic + ESU, which replaces the incremental subgraph matching algorithm in our genetic framework with a widely adopted baseline method, *i.e.*, *ESU* [51]; 2) Random + ESU: search randomly initialized subgraph templates with *ESU*. Besides, we also add a Bayesian Optimization (BO) + ESU baseline, which uses the the classic Bayesian optimization to search for optimal subgraph templates [56]. We use the BO model implemented in Hyperopt framework [4]. The optimization results are shown in Figure 5 and Table 3. Specifically, Table 3 shows the proposed Genetic method (Genetic + INC.) outperform all three baselines across all the datasets. Our genetic framework generates 3.0~8.0% performance gain over the initialized simple edge template within 3000 seconds, which is 0.6~2.8% higher compared with the best performing baseline. Moreover, the optimized performance is comparable with the hand-crafted subgraph templates in Table 1.

Table 3: Classification accuracy (%) and time composition after 3000 seconds genetic optimization.

| | | Hamilton | Lehigh | Rochester | JHU | Amherst |
|---|---|---|---|---|---|---|
| **Classification Accuracy (%)** | Edge Template (Init.) | 25.8 | 23.3 | 22.6 | 26.0 | 28.3 |
| | Random+ESU | 29.2 | 24.5 | 24.8 | 26.0 | 28.3 |
| | BO+ESU | 30.1 | 25.0 | 24.6 | 26.2 | 29.5 |
| | Genetic+ESU | 31.0 | 27.0 | 25.0 | 27.5 | 33.0 |
| | Genetic+INC. | 33.8 | 27.9 | 25.6 | 30.3 | 34.5 |
| **Time Composition (Seconds)** | ESU Matching | 648.7 | 889.4 | 2592.0 | 1389.2 | 1357.1 |
| | INC. Matching | 24.0 | 55.3 | 134.3 | 129.9 | 124.6 |
| | Model Evaluation | 20.9 | 33.7 | 45.0 | 31.5 | 18.1 |

Figure 5 shows a case study of efficiency and searched subgraph templates of Algorithm 1. We observe that Algorithm 1 can generate similar prediction accuracy compared to the hand-craft subgraph templates in reasonable search time. Moreover, these subgraph templates are data-dependent.

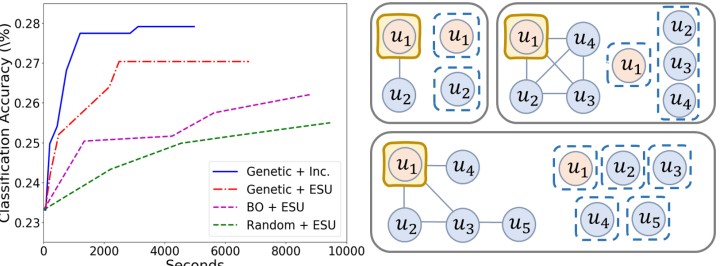

Figure 5: Effectiveness of the proposed genetic algorithm on the *Lehigh* Dataset. Left: search efficiency; right: subgraph templates.

We also evaluate the performance of various hand-craft subgraph templates in Appendix D.4. We can see that the classification accuracy on *Lehigh* dataset ranges from 22.6% to 26.1%, which show the importance of subgraph templates and the need for automatic subgraph template search.

## 5 Conclusion

In this paper, we design a theoretical framework to examine GNN's expressiveness through the lens of Ego-AE. We prove MPNNs have fundamental limitations in capturing this important structural property. Moreover, we propose a provably expressive GNN model, *i.e.*, GRAPE, which effectively extend GNN's capability in modeling structural roles. We also design a genetic subgraph template search algorithm to automatically optimize the model architecture. Experiments on real-world datasets show consistent performance gain of the proposed methods. One potential limitation of our model is the scalability to large-scale real-world graphs (see complexity analysis in Appendix C.3). However, since our model is defined on localized Ego-AE, sampling technique similar to GraphSage [19] can be designed to ensure the feasibility of computation overhead, which we will leave as a future work. Besides, our paper proposes a general GNN framework and it could be customized for specific application domains, *e.g.*, molecular property prediction [50], to achieve additional performance gain.

## Acknowledgments and Disclosure of Funding

This work was supported in part by the National Key Research and Development Program of China under grant 2020AAA0106000, the National Natural Science Foundation of China under U1936217, 61971267, 61972223, 61941117, 61861136003. The authors declare no competing interest.

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
