# A Comparison with Existing GNN Variants

As described in Section 2, there are two directions to augment the expressive power of MPNNs: augmenting node features and designing novel architectures. However, we show in Proposition 3.2 that the important graph property of Ego-AE cannot be captured by the classical MPNN framework, which has not been explored by existing GNN variants. Specifically, previous efforts in feature augmented GNN variants aimed to improve the power by incorporating various additional feature, *e.g.*, graph position [59, 27], spatial orientation of edges [25] and port numbering [42]. However, these additional features are often difficult to generalized [5] and they do not investigate the Ego-AE property. In terms of novel architecture, *GIN* [55] and *k-GNN* [38] were proposed to optimize the expressiveness in graph isomorphism test, which followed the hierarchy of *1-WL* and *k-WL* framework, respectively. Recent work showed that combining multiple aggregator functions can also improve the expressive power [9]. However, these works mainly investigated the expressiveness of GNNs with graph isomorphism test, which is proven to be an easier task than Ego-AE in previous work [46].

Our work follows the later branch of research. Specifically, we propose a novel GNN model, *i.e.*, GRAPE, which can theoretically capture the structural roles defined by Ego-AE. Moreover, we design a genetic algorithm and a compatible incremental subgraph matching algorithm to efficiently search the architecture of GRAPE, which allows it to automatically focus on the most relevant Ego-AE feature in given datasets. To conclude, the proposed GRAPE fundamentally extends GNN's capability in modeling automorphic equivalences and reduces the barrier of generalizing to different datasets.

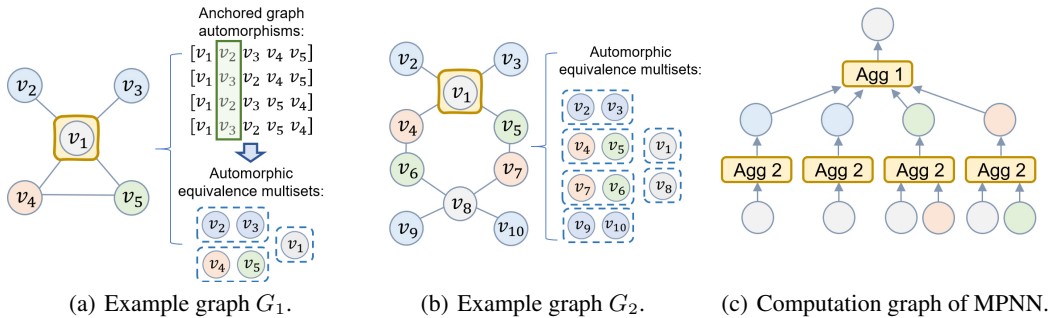

(a) Example graph $G_1$.    (b) Example graph $G_2$.    (c) Computation graph of MPNN.

Figure A1: An illustration of the Ego-AE sets in example graphs and the limitations of current GNNs, where the nodes with same colors have identical features.

# B Proofs

## B.1 Proposition 3.2

*Proof.* We provide a constructive proof for Proposition 3.2 in Figure A1. We can observe that example graphs $G_1$ and $G_2$ have different Ego-AE sets for the corresponding ego node $v_1$, but their computation graphs of 2-layer GNN are exactly the same, which are shown in Figure A1 (c). In fact, since each node in graph $G_2$ can be mapped to a node in graph $G_1$ with identical 1-hop neighborhood, GNNs with arbitrary layers cannot discriminate these two nodes. Therefore, it constitutes a constructive proof for Proposition 3.2.

□

## B.2 Theorem 3.1

*Proof.* Suppose node $v_a$ and $v_b$ have different Ego-AE sets $\mathcal{T}^a = \{\mathcal{A}^a_1, ....\mathcal{A}^a_j, ...\}$ and $\mathcal{T}^b = \{\mathcal{A}^b_1, ..., \mathcal{A}^b_j, ...\}$. Without loss of generality, we assume $\mathcal{A}^a_j$ and $\mathcal{A}^b_j$ are the two sets of nodes that are different. The recently developed "deep set" theory provides a framework for injective functions on set data [60], which is then extended to set scenario showing sum operator is an injective function on set [55]. Therefore, $\mathsf{SUM}(\cdot)$ will map them to distinct embeddings $\boldsymbol{y}^a_i$ and $\boldsymbol{y}^b_j$ since it is an injective function.

**Algorithm A1** GRaph AutormorPhic Equivalent Network (GRAPE)

---

1: Input graph $G = (\mathcal{V}, \mathcal{E})$, node feature $\mathcal{X}(v)$, Ego-AE sets $\{\mathcal{T}_1, ..., \mathcal{T}_L\}$ for $L$ subgraph templates, layer $k \in [1, K_1]$, non-linearity $\sigma(\cdot)$;
2: Node embedding $\boldsymbol{h}(v)$, $\boldsymbol{h}^0(v) \leftarrow \mathcal{X}(v), \forall\, v \in \mathcal{V}$;
3: Ground truth $y(v)$; loss function $Loss(\cdot, \cdot)$; epochs $n \in [1, N]$;
4: **for** $n \in 1, ..., N$ **do**
5:     **for** $k \in 1, ..., K_1$ **do**
6:        *# AE-aware aggregator with various subgraph templates*
7:        **for** $\mathcal{T}_l \in \{\mathcal{T}_1, ..., \mathcal{T}_L\}$ **do**
8:           Compute $\boldsymbol{h}_l^k(v)$ using (2)
9:        **end for**
       *# Squeeze-and-excitation module to fuse multi templates embeddings*
10:        compute $\boldsymbol{\alpha}^k$ using (4);
11:        compute $\boldsymbol{h}^k(v)$ using (3);
12:     **end for**
13:     $\hat{y}(v) = MLP(\boldsymbol{h}^{K_1}(v))$
14:     $Back\_Propagation(Loss(\hat{y}(v), y(v)))$
15: **end for**
16: Return $Accuracy(\hat{y}(v), y(v))$

---

Since node feature $\mathcal{X}$ is countable, the embedding of Ego-AE sets $\boldsymbol{y}$ is also countable. Therefore, it can be mapped to natural numbers with some function $Z : \mathcal{Y} \to \mathbb{N}$. Each node has a set of embeddings corresponding to its Ego-AE sets $Y = \{\mathsf{SUM}(\{\mathcal{X}(v) | v \in \mathcal{A}_j\}) \mid \mathcal{A}_j \in \mathcal{T}\}, Y = \{\boldsymbol{y}_j\} \subset \mathcal{Y}$, where the cardinality of $Y$ is defined by the number of Ego-AE sets $M$ for the given subgraph template. We can construct a function $f(\boldsymbol{y}) = M^{-Z(\boldsymbol{y})}$ so that $\sum_{j \in [1,M]} \beta_j f(\boldsymbol{y}_j), \boldsymbol{y}_j \in Y$ is unique for each set of embeddings, *i.e.*, $\sum_{j \in [1,M]} \beta_j f(\cdot)$ is an injective function on $Y$ [60].

Therefore, for any injective function $g(\cdot)$, the $g(\sum_{j \in [1,M]} \beta_j f(\mathsf{SUM}(\{\mathcal{X}(v) | v \in \mathcal{A}_j\})))$ can learn distinct embedding for $v_a$ and $v_b$, since the composition of three injective functions is still an injective function. If we use $\psi(\cdot)$ to denote $g \circ f$, then it is equivalence to $\psi(\sum_{j \in [1,M]} \beta_j \mathsf{SUM}(\{\mathcal{X}(v) | v \in \mathcal{A}_j\}))$, where $\psi(\cdot)$ is an injective function since both $f(\cdot)$ and $g(\cdot)$ are injective. Therefore, there exist some injective functions $\psi(\cdot)$ that allow the AE-aware aggregator to learn distinct node embedding for $v_a$ and $v_b$. Note that since the initial node feature $\mathcal{X}$ is countable and the AE-aware aggregator is injective, the hidden embeddings $\boldsymbol{h}^{k-1}(v), k \in [2, K]$ is also countable. Therefore, this argument holds for AE-aware aggregators in all hidden layers as described in (2). Besides, the universal approximation theorem suggest that we can use multi-layer perception (MLP) with at least one hidden layer to approximate any injective function. Therefore, we can use $\mathsf{MLP}(\cdot)$ to approximate the injective function $\psi(\cdot)$. As a result, our AE-aware aggregator described in (2) can discriminate the nodes with distinctive Ego-AE feature.

## B.3 Proposition 3.1

*Proof.* As defined in Figure A2 and Appendix C.2, we have two types of mutations: a) *node mutation* that attaches a new node to a randomly selected node in parent subgraph template; and b) *edge mutation* that randomly adds an edge between two unconnected nodes in parent subgraph template.

Given a graph $G = (\mathcal{V}, \mathcal{E})$, let the matched instance set of a parent subgraph template $S_p = (\mathcal{U}_p, \mathcal{R}_p)$ be $\mathcal{M}_p$. We define the mutated children subgraph template as $S_c = (\mathcal{U}_c, \mathcal{R}_c)$. Based on the definitions of *edge mutation* and *node mutation*, the parent subgraph template is a subgraph of the children subgraph template, *i.e.*, $\mathcal{U}_p \subset \mathcal{U}_c, \mathcal{R}_p \subset \mathcal{R}_c$. Therefore, the matched instances of parent subgraph template will be a partial match of the children subgraph template, *i.e.*, $m_p \subset m_c : \exists m_p \in \mathcal{M}_p, \forall m_c \in \mathcal{M}_c$. Therefore, $m_c$ can be efficiently identified by incrementally extending $m_p$. □

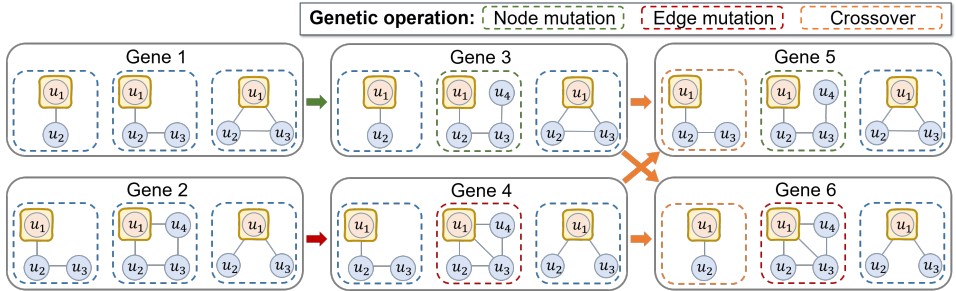

Figure A2: Illustration of the *mutate* and *crossover* operations in the proposed genetic algorithm.

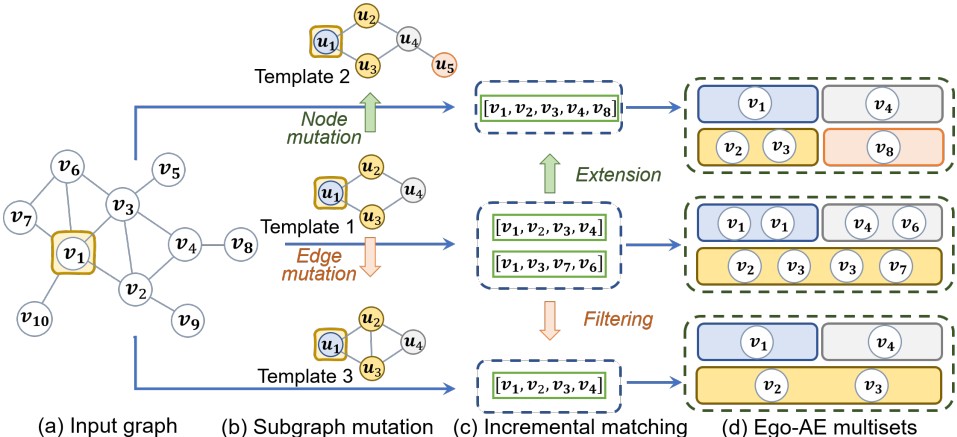

Figure A3: Illustration of identifying Ego-AE sets with incremental subgraph matching.

## C   More Details for Section 3

### C.1   GRAPE Algorithm in Section 3.1

The GRAPE algorithm (Algorithm A1) takes a graph $G$, node feature vector $\mathcal{X}(v)$ and the Ego-AE sets $\{\mathcal{T}_1, ..., \mathcal{T}_L\}$ identified by given subgraph templates as input. In each layer, GRAPE uses AE-aware aggregator to transform the features in each Ego-AE set based on (2). Besides, the embeddings learned from each subgraph template are fused together with a squeeze-and-excitation module based on (3) and (4). Finally, the final layer embedding is transformed by a two-layer multi-layer perception module (MLP) to generate prediction results.

### C.2   Genetic Algorithm in Section 3.2

#### C.2.1   Illustration of Genetic Operations

Figure A2 illustrates the *node mutation*, *edge mutation* and *crossover* operations in the proposed genetic algorithm. Specifically, the *node mutation* will generate a *children subgraph* by randomly adding one node to the input *parent subgraph*, while *edge mutation* generates a *children subgraph* by randomly an edge between two unconnected nodes in the input *parent subgraph*. Besides, the *crossover* operation will randomly exchange some subgraph templates between two genes. These operations effectively allow us to gradually search for slightly more complicated subgraph templates and try out different combinations of subgraph templates.

#### C.2.2   Incremental Subgraph Matching

To accelerate the matching from the subgraph template to each node's neighborhood, we propose to leverage the similarity between *children subgraph* and *parent subgraph*. Inspired by Proposition 3.1, we design an incremental subgraph matching algorithm that identifies the matched instances of

*children subgraph* by only examining the matched instances of *parent subgraph*, which is illustrated in Figure A3. Specifically, given the matched instances of *Template 1*, *i.e.*, $[v_1, v_2, v_3, v_4]$ and $[v_1, v_3, v_7, v_6]$, we identify the matched instances of its node mutation children *Template 2* by exploring the neighbors of the current matched instances. Since the newly added node is attached to $u_4$, we will only examine the neighbors of nodes that mapped to $u_4$, *i.e.*, $v_6$ and $v_4$. Therefore, we find $v_8$ as a feasible candidate and identify the matched instance of *Template 2* as $[v_1, v_2, v_3, v_4, v_8]$. As for the edge mutation children *Template 3*, we only need to examine the newly added edge between $u_2$ and $u_3$ in the matched instances. We find $v_2$ and $v_3$ are indeed connected but there is no edge between $v_3$ and $v_7$. Therefore, we identify one matched instance for *Template 3*, *i.e.*, $[v_1, v_2, v_3, v_4]$. Complex subgraph templates with numerous nodes usually result in exponential growth in matching computation time compared to the simpler ones [8], but they often have much fewer matched instances. Therefore, by leveraging the feature of genetic search with incremental subgraph matching, we can significantly reduce the computation complexity by only examining the matched instances of parent subgraph instead of starting from scratch.

### C.3 Time Complexity

**GRAPE model.** Here, we analyze time complexity of one forward pass of GRAPE model. Specifically, suppose we have $L$ subgraph templates, each template has $M$ Ego-AE sets and each set contains $Q$ nodes on average, the overall time complexity of training GRAPE with $K$ layers is $\mathcal{O}(|\mathcal{V}|LMQK)$, where $|\mathcal{V}|$ is the number of nodes on graph. Empirically, the matched neighbor of each subgraph is a subset of each node's neighborhood, *i.e.*, $MQ \leq |\mathcal{N}|$. Therefore, GRAPE's time complexity is comparable to the popular MPNNs, *e.g.*, GraphSAGE and GCN, which typically have a time complexity of $\mathcal{O}(|\mathcal{V}||\mathcal{N}|K)$.

**Incremental subgraph matching.** We assume the *parent subgraph* has $\Pi_e$ ego-centered automorphisms and $|\mathcal{M}_p|$ match instances, and each node has $|\mathcal{N}|$ neighbors on the target graph. Then, the complexity of identifying the match instances after adding node *mutation* is $O(|\mathcal{M}_p||\mathcal{N}|\Pi_e)$, since we only need to examine the possible extensions. Similarly, the complexity of examining adding edge *mutation* is $O(|\mathcal{M}_p|\Pi_e)$. They both have significantly lower the $\mathcal{O}(|\mathcal{V}|!|\mathcal{V}|)$ worst case computation complexity deduced by previous work [8]. The average case computation complexity can not be analytically estimated unless very restrictive assumptions are made. However, the empirical experiments in Table 3 and Figure 5 demonstrate our proposed incremental search algorithm can significantly outperform baselines on real-world datasets. Moreover, similar sampling approach as in GraphSAGE [19] can be adopted to control the size of match instance set $|\mathcal{M}_p|$, which can ensure the computational footprint of our algorithm is feasible.

## D    Experiments Details

### D.1    Experiment Setting and Hyper-parameter

Following the setting in previous works [55], we perform a grid search on the following hyper-parameters: 1) embedding size $\in \{16, 32\}$; 2) the dropout rate $\in \{0.3, 0.5\}$; 3) L2 regularization coefficient $\in \{3 \cdot 10^{-5}, 5 \cdot 10^{-5}\}$; 4) initial learning rate $\in \{0.01, 0.03\}$, which is decayed by 50% for every 100 epochs. To improve the robustness of experiment results, we report the average and standard deviation of each model's performance over 10 runs. In each run, we randomly split the datasets into 60% training set, 20% validation set and 20% test set. Specifically, we use the training set to learn the models, and report the classification performance o test set. We train each model for 500 epochs with early stopping of 50 window size, *i.e.* the training is terminated if the model's performance on validation set does not improve for consecutive 50 epochs. Our model and all the baseline models are implemented in Pytorch [39] with the Adam optimizer. We evaluate them on a single machine with 4 NVIDIA GeoForce RTX 2080 GPUs.

### D.2    Subgraph Templates Design

We design multiple subgraph templates to allow GRAPE to capture various automorphic equivalences, which are presented in Figure A4. Here, we discuss the motivations for their design.

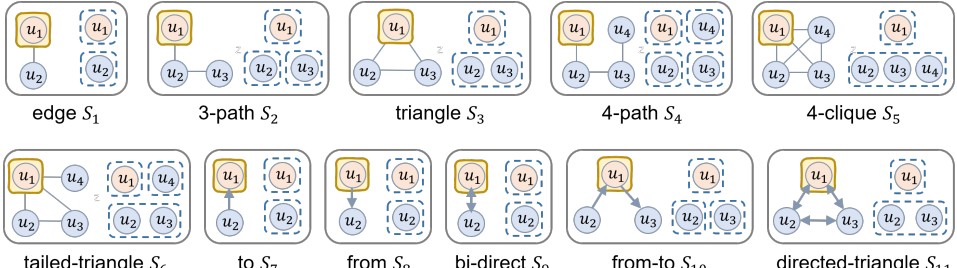

Figure A4: Illustration of the designed subgraph templates.

1) *Edge* $S_1$: it captures the basic connection in graph data, and has two sets of Ego-AE nodes, *i.e.*, $\{u_1\}$ and $\{u_2\}$. Therefore, it leads to two Ego-AE sets: one contains the ego node itself (corresponding to $u_1$) and the other contains all the 1-hop neighbors (corresponding to $u_2$).

2) 3-path $S_2$: it captures the 2-hop neighborhood of the ego node and partition the neighbors into three Ego-AE sets based on their hops from ego node, which maps to $\{u_1\}$, $\{u_2\}$ and $\{u_3\}$, respectively. In the context of citation network, it captures the documents that co-cite one document. Besides, it captures the individuals that have common friends in social network.

3) Triangle $S_3$: This template captures an important pattern in graph data, *i.e.*, triangle. Based on the triadic closure theory [23, 20], this template captures the strong ties in social graph, *i.e.*, the individuals that form triangle structure tend to have similar feeling about an object. This template maps the neighborhood into two Ego-AE sets that corresponds to $\{u_1\}$ and $\{u_2, u_3\}$, respectively. But differs from edge template $S_1$, $\{u_2, u_3\}$ maps to the neighbors that tend to have stronger influence on the ego node.

4) 4-path $S_4$: similar to the 3-path template, this template maps the nodes in 3-hop neighborhood into 4 Ego-AE sets based on their hops from ego node. It allows the model to access more far away features.

5) 4-clique $S_5$: this template captures the closed connected communities in graph data, which tend to exhibit the "homophily effect" [36]. It maps the neighborhood into two Ego-AE sets that corresponds to $\{u_1\}$ and $\{u_2, u_3, u_4\}$, respectively.

6) Tailed-triangle $S_6$: on the basis of triangle template $S_3$, this template adds an additional neighbor to the ego node. Therefore, it partitions the neighborhood into three Ego-AE sets, *i.e.*, $\{u_1\}$, $\{u_2, u_3\}$ and $\{u_4\}$, where $\{u_2, u_3\}$ identifies the neighbors with strong ties and $\{u_4\}$ identifies the neighbors connected by simple edge.

7) To $S_7$: it identifies the neighbors that point to the ego node in directed graph. In the context of e-commerce co-purchase network, $u_2$ maps to items that often lead to the purchase of $u_1$.

8) From $S_8$: it identifies the neighbors that have directed edges from the ego node. In the context of e-commerce co-purchase network, the purchase of ego node $u_1$ often leads to the purchase of $u_2$.

9) Bi-direct $S_9$: it identifies the neighbors that are connected to and from the ego node, which are the intersection of the nodes identified by template $S_7$ and $S_8$. Therefore, $u_2$ maps to the items that are frequently co-purchased with $S_{11}$.

10) From-to $S_{10}$: it maps to the unions of the nodes identified by template $S_7$ and $S_8$. Specifically, it partitions the neighborhood into three Ego-AE sets, *i.e.* $\{u_1\}$, $\{u_2\}$ and $\{u_3\}$, which correspond to the ego node itself, the nodes that point to ego node, and the nodes that are pointed from ego node, respectively.

11) Directed-triangle $S_{11}$: similar with the triangle template $S_3$, this template captures the triangle patterns in directed graph setting. Specifically, it leads to two Ego-AE sets, which correspond to $\{u_1\}$ and $\{u_2, u_3\}$, respectively.

Specifically, we use $\{S_1, S_2, S_3, S_4, S_6\}$ for citation datasets, $\{S_1, S_2, S_3, S_5, S_6\}$ for social datasets, and $\{S_7, S_8, S_9, S_{10}, S_{11}\}$ for amazon dataset.

## D.3 Results with Dummy and Random Initialized Node Feature

Here, we present the model performance on datasets with dummy and random initialized node feature in Table A1 and Table A2, where the original node feature vectors are replaced by all-ones vectors and randomly generated vectors. We can observe that GRAPE consistently outperforms all baseline models with both experiment settings, where the relative accuracy gain over the best baseline models reaches up to 44.7% and 48.6%, respectively. It indicates GRAPE is expressive with or without node feature, which showcases its capacity in capturing rich structural features.

Table A1: Classification accuracy on datasets with dummy node feature (%). The best-performing GNNs are in boldface.

| | Social | | | | | Citation | | Ecomm. |
|---|---|---|---|---|---|---|---|---|
| Model | Hamilton | Lehigh | Rochester | JHU | Amherst | Cora | Citeseer | Amazon |
| GCN | 19.5±1.5 | 23.4±1.3 | 22.4±1.6 | 19.3±0.8 | 18.1±1.7 | 31.0±1.2 | 21.5±0.9 | 38.8±1.0 |
| GraphSAGE | 18.8±3.3 | 20.6±3.1 | 20.4±2.3 | 18.6±2.2 | 17.0±2.4 | 29.7±1.6 | 20.0±0.9 | 38.5±1.0 |
| GIN | 22.7±5.1 | 19.2±2.5 | 22.1±1.7 | 24.2±4.0 | 18.9±3.9 | 29.5±1.3 | 21.1±1.5 | 39.1±0.9 |
| GAT | 16.8±1.4 | 23.5±3.4 | 21.5±0.9 | 17.6±1.1 | 16.9±2.3 | 25.8±2.9 | 18.2±0.9 | 38.7±1.2 |
| Geniepath | 27.5±2.8 | 23.3±1.7 | 21.7±1.5 | 21.4±3.6 | 25.3±3.4 | 31.4±1.7 | 19.1±1.0 | 38.2±0.9 |
| Meta-GNN | 23.7±1.6 | 25.6±1.5 | 25.8±1.1 | 28.8±2.3 | 23.1±3.2 | 30.4±1.4 | 24.5±1.7 | 38.6±1.1 |
| Mixhop | 19.8±0.0 | 23.1±0.1 | 17.9±0.1 | 18.6±0.2 | 17.3±0.1 | 31.9±0.1 | 18.1±0.0 | 38.9±0.1 |
| DE-GNN | 21.7±2.1 | 24.7±2.2 | 18.0±0.0 | 18.3±0.1 | 18.6±2.2 | 31.8±0.1 | 17.9±0.3 | 38.9±0.0 |
| GRAPE | **39.8±3.8** | **28.9±2.4** | **32.5±1.4** | **35.8±2.3** | **36.6±3.2** | **34.4±3.3** | **26.3±0.8** | **42.9±0.7** |

Table A2: Classification accuracy on datasets with random initialized node feature (%). The best-performing GNNs are in boldface.

| | Social | | | | | Citation | | Ecomm. |
|---|---|---|---|---|---|---|---|---|
| Model | Hamilton | Lehigh | Rochester | JHU | Amherst | Cora | Citeseer | Amazon |
| GCN | 19.7±3.3 | 23.2±0.1 | 21.9±0.8 | 19.0±2.4 | 17.2±1.2 | 39.9±9.8 | 29.6±9.6 | 38.9±0.1 |
| GraphSAGE | 17.3±5.6 | 15.2±8.2 | 17.8±6.2 | 15.4±7.9 | 15.0±5.9 | 34.9±0.7 | 23.0±2.4 | 35.9±1.1 |
| GIN | 35.7±2.4 | 24.3±2.1 | 32.1±1.6 | 32.4±3.0 | 30.0±5.4 | 27.3±0.1 | 20.0±0.0 | 37.0±0.2 |
| GAT | 17.6±3.4 | 24.3±2.1 | 21.9±1.8 | 18.0±3.4 | 16.2±1.4 | 25.1±2.2 | 19.2±1.7 | 38.6±0.6 |
| Geniepath | 28.1±3.1 | 23.3±2.1 | 21.4±0.6 | 22.1±4.3 | 24.9±3.6 | 31.2±7.2 | 18.0±3.5 | 38.0±1.2 |
| Meta-GNN | 24.7±2.6 | 25.1±1.8 | 26.0±2.2 | 28.9±3.1 | 23.6±2.8 | 30.1±1.6 | 25.3±0.7 | 38.4±0.7 |
| Mixhop | 21.3±0.2 | 24.1±1.3 | 18.4±0.3 | 17.6±0.4 | 18.5±0.9 | 31.8±0.5 | 19.8±1.2 | 38.7±0.4 |
| DE-GNN | 21.9±1.6 | 23.2±0.1 | 18.0±0.0 | 21.6±1.0 | 20.7±3.3 | 31.9±0.0 | 18.2±0.1 | 38.9±0.0 |
| GRAPE | **38.9±3.0** | **31.1±1.5** | **34.3±0.6** | **34.0±2.0** | **37.2±3.5** | **40.7±3.5** | **44.0±4.7** | **39.1±3.5** |

## D.4 Results with Different Subgraph Templates

Table A3: Classification accuracy with different subgraph templates on *Lehigh* dataset (%).

| | Subgraph Templates | | | | | |
|---|---|---|---|---|---|---|
| Model | $S1$ | $S2$ | $S3$ | $S4$ | $S5$ | $S6$ |
| GRAPE | 23.3±1.7 | 22.6±1.3 | 26.1±4.0 | 22.9±0.6 | 23.2±1.3 | 23.3±0.9 |

## D.5 License of Assets

The source code will be shared under MIT license. All the datasets used in this research is public available.