# OpenReview forum: "Automorphic Equivalence-aware Graph Neural Network"
_NeurIPS.cc/2021/Conference — NeurIPS 2021 Poster_

### Official Review · Reviewer_nRJR · 2021-07-14

**Rating:** 6
**Confidence:** 2

**Summary:**


This paper studies the problem of improving the expressive power of GNN models. It computes the automorphism equivalence set for each node and uses the set to enhance the message passing. This method is proven to be more expressive than standard MPNN.


**Limitations And Societal Impact:**

The authors addressed the limitations and potential negative societal impact.

**Main Review:**

### Originality

This paper studies the problem of improving the expressive power of GNN models. It proposes to use automorphism-aware aggregation to achieve it.

Using a genetic algorithm to search for motif templates is novel.

The method using automorphism equivalence matching and counting to improve the expressive power of GNNs is previously proposed in [6], while the relation of this work to [6] is not adequately addressed and discussed. [6] also proved by matching and counting subgraph isomorphism, the expressive power of GNNs can be improved. Though the AE-aware aggregator is not the same as the counting based aggregation in [6], the similarity in the core idea should definitely be discussed more.


### Quality

The claims in this paper are well justified. The experiments are well designed and able to show the effectiveness of the method.

### Clarity

The presentation is clear.

### Significance

This paper proposes to use automorphic matching to improve the graph neural networks’ expressive power. The motif template searching genetic algorithm can be useful in this direction, while the relation to [6] is not clearly discussed. My major concern is about the novelty here. I may raise my score if the connection to [6] can be clarified.



**Time Spent Reviewing:**

5

---

### Official Review · Reviewer_ULms · 2021-07-16

**Rating:** 6
**Confidence:** 3

**Summary:**

The paper proposes a new GNN architecture, named GRAPE, which aims to take into account automorphic equivalence relationships in the egonets surrounding a node to improve the discriminative power of the learning architecture.
The authors present two theorems to support their architectural design and show numerical experiments on a number of datasets in which the new architecture appears to perform well.

**Limitations And Societal Impact:**

Given the broad literature on limitations of GNN (in terms of WL test, robustness guarantees, etc), I find it surprising that a discussion of the weaknesses of the presented method is not really present.
The possible societal impact of using graph neural networks in classification task (e.g., in social networks, as considered here) is also not discussed in any detail.

**Main Review:**

The paper considers a variation of GNN which account for (local) AE relationships. In my view, there are already many related ideas (though not quite the same) out there, so I would rank the originality of this approach to be not that high.

I found a number of claims made in the paper to be somewhat surprising or lacking precision, e.g., it is stated that there are efficient algorithms to solve the graph isomorphism / automorphism problem. I don't think this is the case. These are clever heuristcs, but as far as I am aware there are no efficient (polynomial time) algorithms for solving the isomorphism problem.

In a similar vein I found the discussion related to the WL algorithm confusing -- the WL algorithm is precisely used in the context of graph isomorphisms, and previous works (Morris et al, Xu et al) show that standard message passing cannot be more powerful than this algorithm, but can in fact be as powerful as the WL algorithm. However, the WL algorithm can distinguish non-automorphic nodes based on their neighborhoods; so I am not fully sure I get the point of Proposition 3.1 and Theoem 3.2.
As another point, the description of the genetic algorithm to be lacking sufficent detail to appreciate what is really done here.

Overall I found the theoretical contribution here to be not that convincing. In my view the proposed architecture is largely of interest because of its empirical performance -- which appears to be good. However, in this context a somewhat more rigorous experimental setup would have made for a much stronger paper and I would urge the authors to improve this section by including more relevant comparisons (like GNNs with random/unique initial node features, or GNNs based on higher-order WL equivalents); likewise the paper would benefit from the inclusion of a larger array of datasets.

============
Update post author reponse/discussion. I have raised my score to 6 following the responses of the authors

**Time Spent Reviewing:**

3

---

### Official Review · Reviewer_jGmr · 2021-07-17

**Rating:** 6
**Confidence:** 4

**Summary:**

In this paper, the author proposed a GNN model that leverages the concept of automorphic equivalence. Specifically, they proposed GRAPE that uses learnable AE-aware aggregators to explicitly differentiate the structural equivalences of each node's neighbors with the aids of various subgraph templates. Moreover, the author theoretically prove that GRAPE is expressive in terms of generating distinct representations for nodes with different AE features. Experiment on real-world dataset proves the effectiveness of GRAPE.

**Limitations And Societal Impact:**

To me, there is no obvious negative societal impact.

**Main Review:**

Overall, the paper is well written, and the formulation process including theocratic proofs are adequate and solid. However, I have concerns of the practicability of the proposed method. I think it is a borderline paper between 5-6 (weak reject to weak accept).

Reasons to accept:
1. The paper is well written.
2. The formulation process including theocratic proofs are solid.

Reasons to reject:
1. The proposed algorithm requires subgraph template, which may not be easily available in many real-world applications. In addition, I didn't find the studies for the impact of different subgraph templates to the performance of the proposed algorithm. Does different subgraph templates and different number of templates have a huge impact to the proposed algorithm?
2. The proposed algorithm seems to increase the complexity and might not be easily applied to real world applications. Although the authors showed the comparison of different algorithm's training time in Figure 4, the authors mentioned that two baselines Geniepath and GAT are trained on CPU, which is not a fair comparison. It is true that GAT used attention mechanism which will cost more GPU memory, but you can modify it to batch training, like in [1] https://arxiv.org/pdf/1806.01973.pdf, rather than training the whole graph at once.

**Time Spent Reviewing:**

2

---

### Official Review · Reviewer_H8ZW · 2021-07-18

**Rating:** 7
**Confidence:** 4

**Summary:**

Overall this work contributes novel ideas from multiple aspects from introducing the new concept of GNNs being automorphic equivalence-aware to further push the expressiveness of these models, including the developed local definition of how to integrate this into the model, to the development of a genetic algorithm-based method for avoiding hand-crafted/hyperparameter tuned subgraph templates that are instead learned/efficiently searched.

**Ethical Concerns:**

No ethical concerns.

**Limitations And Societal Impact:**

This is not needed in the body of their work and they have provided the following in their checklist:
"""
Did you discuss any potential negative societal impacts of your work?
Since our paper propose a general graph neural network model and the datasets we use are public available, no direct potential negative societal impacts is associated with our work.
"""

**Main Review:**

Strengths:

S1) See the summary above.

S2) This work not only presents an empirically well performing method, but also theoretically proves the expressiveness to learn distinct representations of nodes having different local automophic equivalence sets which they have defined.

S3) The background/related work of this work is well presented to provide context for the later method developed.

S4) The empirical evaluation is quite thorough. For example, even an investigation is made on the strength of the proposed GA-based subgraph search algorithm against other search algorithms such as randomized and Bayesian optimization-based method where their proposed method consistently outperforms the others across all datasets.

Weaknesses:

W2) The motivation of the structural equivalence in the second paragraph of the introduction is still not very convincing from a high-level when compared with the more traditional message passing GNN framework that also includes more information from those nodes in a clique formation as compared to longer-reaching weak ties. Although this is presented better throughout the paper, at this stage it still seems lacking.

W2) The comment that “previous analytic methods only classify nodes into categorical equivalence sets, which cannot be optimized for various specific applications in an end-to-end manner” seems to be incorrect as such non-categorical (structure) information has been presented in numerous self-supervised learning approaches to help guide graph neural networks. Please see the following surveys [1,2,3].

Other comments:

1) The way this work is searching for subgraph templates is somewhat related to the work in 3D GNNs that are learning appropriate 3D kernels [4] in point cloud and 3D molecule domain, which might be considered as related work.

2) It would be appreciated to have more discussion on some of the findings (although acknowledging that space is already an issue with this work with much content being moved to the supplemental file), such as that in Figure 4(b) where the first layer is extremely biased towards S1, but in the second layer there seems to be hardly any statistical difference between the subgraphs. Also, discussing subgraph templates such as the triangle S3 in the main work, but only defining in the supplemental is not preferred.


References:

[1] Xie, Yaochen, et al. "Self-supervised learning of graph neural networks: A unified review." arXiv preprint arXiv:2102.10757 (2021).

[2] Liu, Yixin, et al. "Graph self-supervised learning: A survey." arXiv preprint arXiv:2103.00111 (2021).

[3] Wu, Lirong, et al. "Self-supervised on Graphs: Contrastive, Generative, or Predictive." arXiv preprint arXiv:2105.07342 (2021).

[4] Lin, Zhi-Hao, Sheng Yu Huang, and Yu-Chiang Frank Wang. "Learning of 3D Graph Convolution Networks for Point Cloud Analysis." IEEE Transactions on Pattern Analysis and Machine Intelligence (2021).

I have read through the other reviews and responses, but am not inclined to update my score. Thank you.

**Time Spent Reviewing:**

4

---

### Author Response · Authors · 2021-08-10
**Response to Reviewer H8ZW**

**Q1.** *The motivation of the structural equivalence in the second paragraph of the introduction is still not very convincing from a high-level when compared with the more traditional message passing GNN framework that also includes more information from those nodes in a clique formation as compared to longer-reaching weak ties. Although this is presented better throughout the paper, at this stage it still seems lacking.*

**Response:** Thanks for the constructive suggestions. Indeed, the motivation is a bit lacking in the beginning of our paper. We will strengthen it with concrete examples in the final version.

**Q2.** *The comment that “previous analytic methods only classify nodes into categorical equivalence sets, which cannot be optimized for various specific applications in an end-to-end manner” seems to be incorrect as such non-categorical (structure) information has been presented in numerous self-supervised learning approaches to help guide graph neural networks. Please see the following surveys [1,2,3].*

**Response:** Thanks for the suggestions. We agree that the statement is inappropriate because the recent self-supervised research use structural feature to guide GNNs. We will clarify this misleading statement and explore our model's connection with the self-supervised research in the final version.

**Q3.** *The way this work is searching for subgraph templates is somewhat related to the work in 3D GNNs that are learning appropriate 3D kernels [4] in point cloud and 3D molecule domain, which might be considered as related work.*

**Response:** Thanks for the suggestion. We agree that paper [4] is closely related in high-level ideas. We believe the most significant differences will be our model is defined on non-euclidean graph data and the subgraph template we used are not differentiable. Following your suggestion, we will discuss our connection to this paper in the final version.

**Q4.** *It would be appreciated to have more discussion on some of the findings (although acknowledging that space is already an issue with this work with much content being moved to the supplemental file), such as that in Figure 4(b) where the first layer is extremely biased towards S1, but in the second layer there seems to be hardly any statistical difference between the subgraphs. Also, discussing subgraph templates such as the triangle S3 in the main work, but only defining in the supplemental is not preferred.*

**Response:** Thanks for the feedback. We will try to better organize our paper and add in more finding analysis in the final version.

---

> ### Comment · Reviewer_H8ZW · 2021-09-03
> **Response to authors**
>
> Thank you for addressing these. I note however that I was already positive about this work and am not inclined to raise/lower my score based on your response (or after reading through the other reviews/responses).

---

### Author Response · Authors · 2021-08-10
**Response to Reviewer jGmr**

**Q1.** *The proposed algorithm requires subgraph template, which may not be easily available in many real-world applications. In addition, I didn't find the studies for the impact of different subgraph templates to the performance of the proposed algorithm.*

**Response:** Indeed, choosing optimal subgraph templates for different datasets often require domain knowledge and it might have huge impact on the model performance.

- To address this problem, in Section 3.2, we propose a genetic optimization algorithm to automatically and efficiently search for optimal subgraph templates for given datasets.
- As we show in Table 3 and Figure 6, our proposed genetic algorithm can achieve 8% performance gain in accuracy without any predefined subgraph templates.

Therefore, choosing different subgraph templates does significantly impact model performance. More importantly, our proposed genetic algorithm can effectively reduce the barrier of hand-craft templates with reasonable time expense, which makes it applicable in real-world problems. Thanks for your feedback. We will clarify this issue in the final version.

**Q2.** *Does different subgraph templates and different number of templates have a huge impact to the proposed algorithm?*

**Response:** The number of subgraph templates will impact GRAPE's performance. As shown in the below table, the model performance generally increases with the number of subgraph templates, which is automatically searched by the genetic algorithm for 3000 seconds. However, when there are more than 3 subgraph templates, the performance gain of adding new subgraph templates become marginal. We will add in these results to clarify this problem in the final version.

| \# of template | Hamilton | Lehigh | Rochester | JHU  | Amherst |
| :------------: | :------: | :----: | :-------: | :--: | :-----: |
|       1        |   27.6   |  25.2  |   23.8    | 26.5 |  26.3   |
|       2        |   31.4   |  26.1  |   24.2    | 29.2 |  31.8   |
|       3        |   33.8   |  27.9  |   25.6    | 30.3 |  34.5   |
|       4        |   33.9   |  28.1  |   26.0    | 30.8 |  34.9   |
|       5        |   34.0   |  28.1  |   26.3    | 31.6 |  34.6   |

**Q3.** *The proposed algorithm seems to increase the complexity and might not be easily applied to real world applications. Although the authors showed the comparison of different algorithm's training time in Figure 4,the authors mentioned that two baselines Geniepath and GAT are trained on CPU, which is not a fair comparison. It is true that GAT used attention mechanism which will cost more GPU memory, but you can modify it to batch training, like in [1] https://arxiv.org/pdf/1806.01973.pdf , rather than training the whole graph at once.(https://arxiv.org/pdf/1806.01973.pdf)*

**Response:**

1. Indeed, GAT can be optimized with batch training to fit in GPU memory. However, it is more difficult to implement batch version Geniepath, because it uses LSTM to dyanmically determine neighborhood size. We show the training time of batch version GAT in the below table. Comparing with Figure 4 (a), we can see GAT, Mixhop, GraphSage and our GRAPE have similar training time. Therefore, our GRAPE model in practice has comparable complexity as the classic GCN variants, e.g., GraphSage and GAT.

| Time (s) |  1   |   3   |  10   |  30   | 100  | 300  |
| :------: | :--: | :---: | :---: | :---: | :--: | :--: |
|   GAT    |  1   | 0.076 | 0.043 | 0.034 |  0   |  0   |

2. Similar with many GNN variants, the complexity of our model will increase with the size of the training graphs (see complexity analysis in Appendix C.3), which might present difficulty in large scale real-world application. However, since we base our model on a localized automorphism concept (see Definition 3.1), we can sample the neighborhood of each node in the process of incremental subgraph matching (see Appendix C.2.2). That is we can limit the number of match instances starting from each node and early stop the matching when sufficient instances are found. Therefore, we can leverage a similar sampling technique as GraphSage to ensure the memory and computation overhead of our model is feasible to large-scale applications. Thanks for the feedback. We will add the discussion of neighborhood sampling and feasibility to real-world application in the final version as future work.

---

> ### Comment · Reviewer_jGmr · 2021-09-10
> **Response to authors**
>
> Thanks for the response. It answers some of my questions. However, I still decided to hold my original evaluation for this paper.

---

### Author Response · Authors · 2021-08-10
**Response to Reviewer ULms (Part I)**

**First, we'd like to clarify the difference between automorphic equivalence and neighborhood isomorphism.**

Graph automorphism is a special isomorphism that maps a graph to itself. Therefore, automorphic equivalence (*i.e.*, whether two nodes can be mapped to each other with automorphic permutation) is a stricter structural condition than neighborhood isomorphic (*i.e.*, whether two nodes have isomporhic neighborhoods). That is **two automorphically equivalent nodes are always neighborhood isomporphic, while the converse statement is false**. As a result, previous works following the hierarchy of WL test, *e.g.*, GIN, aim to differentiate neighborhood isomorphic nodes, but they cannot capture automorphic equivalence feature. Our GRAPE model aims to fill in this gap.

**Q1.** *I found a number of claims made in the paper to be somewhat surprising or lacking precision, e.g., it is stated that there are efficient algorithms to solve the graph isomorphism / automorphism problem. I don't think this is the case. These are clever heuristics, but as far as I am aware there are no efficient (polynomial time) algorithms for solving the isomorphism problem.*

**Response:** Sorry for the misunderstanding.

- Indeed, there is no polynomial time algorithms for graph isomorphism problem. What we meant *efficient* is empirical not theoretical.
- Specifically, we use the *Nauty* algorithm proposed by McKay [38], which is a clever heuristcs that typically takes less than 1 minute on laptop to enumerate all the automorphisms for graphs less than 100 nodes.
- More importantly, we only need to enumerate the automorphisms of the subgraph templates for once, which typically contains less than 20 nodes to remain semantic-aware and prevent over-fitting.

Therefore, solving the automorphism problem in our model is not a bottleneck in practice. Thanks for the feedback. We will clarify the misleading statement in final version.

>[38] Brendan D McKay and Adolfo Piperno. Practical graph isomorphism, ii. Journal of Symbolic Computation, 60:94–112, 2014.

**Q2.** *In a similar vein I found the discussion related to the WL algorithm confusing -- the WL algorithm is precisely used in the context of graph isomorphisms, and previous works (Morris et al, Xu et al) show that standard message passing cannot be more powerful than this algorithm, but can in fact be as powerful as the WL algorithm. However, the WL algorithm can distinguish non-automorphic nodes based on their neighborhoods; so I am not fully sure I get the point of Proposition 3.1 and Theorem 3.2.*

**Response:** Please first check "the difference between automorphic equivalence and neighborhood isomorphism" at the beginning of this reply.

- WL algorithm is designed to distinguish graph isomorphisms. However, automorphic equivalence (AE) is a more rigorous structural feature than having isomorphic neighborhood, because isomorphism does not need to map the nodes onto themselves. Specifically, two AE nodes always have isomorphic neighborhood, while the nodes with isomorphic neighborhood are not necessary AE [15]. Counter examples are constructed in Ref [S1]: page 154 - 164.
- Therefore, since MPNN is at most as powerful as WL algorithm and WL algorithm is not sufficient to distinguish non-automorphic nodes, MPNN is not expressive enough to capture AE features (Proposition 3.2) while our model fills in this gap (Theorem 3.1).

>[15] Martin G Everett, John P Boyd, and Stephen P Borgatti. Ego-centered and local roles: A graph theoretic approach. Journal of Mathematical Sociology, 15(3-4):163–172, 1990.
>
>[S1] Biggs N, Biggs N L, Norman B. Algebraic graph theory[M]. Cambridge university press, 1993.

**Q3.** *As another point, the description of the genetic algorithm to be lacking sufficient detail to appreciate what is really done here.*

**Response:**
Thanks for the feedback. The genetic algorithm is presented as follow:

- We first introduce this motivation and high-level idea in the Abstract (line 8 - 10) and Introduction (line 57 - 62).
- Then, we lay out the classic framework of genetic algorithm in Related Work (see Section 2.3 Genetic Algorithm).
- After that, we describe all the essential parts of our genetic algorithm with Figure 3 and Section 3.2 (line 198 - 237).
- And, we give Proposition 3.1 to explain why our genetic algorithm will be efficient in theory, which is empirically validated with the experiments in Section 4.3 (see Table 3 and Figure 6).

Due to the limit of space, we leave the further details in Appendix C. We will try to include more details in the final version.

**Q4.** *Overall I found the theoretical contribution here to be not that convincing. In my view the proposed architecture is largely of interest because of its empirical performance -- which appears to be good. However, in this context a somewhat more rigorous experimental setup would have made for a much stronger paper and I would urge the authors to improve this section by including more relevant comparisons (like GNNs with random/unique initial node features, or GNNs based on higher-order WL equivalents).*

**Response:** Thanks for the constructive suggestions.

1. Our paper proposes a more powerful GNN architecture, which is orthogonal to the feature augmentation methods, *i.e.*, the augmented features can also be used in our model. Following your suggestion, we supplement a new experiment with the randomly initiated/unique node features proposed in Ref [S2]. We observe that our proposed GRAPE model consistently outperforms all baselines. Besides, Table 4 and Table 1 show our GRAPE model also outperforms baselines with original and dummy node feature. These results show that our model can achieve performance gain independent of node features.
2. Mixhop, Meta-GNN and De-GNN are the baseline GNN variants that leverage high-order connectivity. More importantly, De-GNN is proven to be more powerful than 1-WL [32]. Experiments show our model can outperform these higher-order GNN variants (see Table 4 and Table 1). Besdies, please note that k-WL (k>1) requires to aggregate k-tuples of nodes, which reduces the sparsity of graph structure and damages their scalability. Previous attempts mostly focus on graph-level tasks with at most hundreds of nodes [S3, S4, S5], which are not comparable to our model. Besides, these works still follow the hierarchy of WL test, and hence they cannot capture AE feature in theory.

**Classification accuracy (%) with random initialized/unique node feature:**

|           | Hamilton | Lehigh | Rochester | JHU  | Amherst | Cora | Citeseer | Amazon |
| :-------: | :------: | :----: | :-------: | :--: | :-----: | :--: | :------: | :----: |
|    GCN    |   19.7   |  23.2  |   21.9    | 19.0 |  17.2   | 39.9 |   29.6   |  38.9  |
| GraphSAGE |   17.3   |  15.2  |   17.8    | 15.4 |  15.0   | 34.9 |   23.0   |  35.9  |
|    GIN    |   35.7   |  24.3  |   32.1    | 32.4 |  30.0   | 27.3 |   20.0   |  37.0  |
|    GAT    |   17.6   |  24.3  |   21.9    | 18.0 |  16.2   | 25.1 |   19.2   |  38.6  |
| Geniepath |   28.1   |  23.3  |   21.4    | 22.1 |  24.9   | 31.2 |   18.0   |  38.0  |
| Meta-GNN  |   24.7   |  25.1  |   26.0    | 28.9 |  23.6   | 30.1 |   25.3   |  38.4  |
|  Mixhop   |   21.3   |  24.1  |   18.4    | 17.6 |  18.5   | 31.8 |   19.8   |  38.7  |
|  De-GNN   |   21.9   |  23.2  |   18.0    | 21.6 |  20.7   | 31.9 |   18.2   |  38.9  |
|   GRAPE   |   38.9   |  31.1  |   34.3    | 34.0 |  37.2   | 40.7 |   44.0   |  39.6  |

> [S2] Sato R, Yamada M, Kashima H. Random features strengthen graph neural networks[C]//Proceedings of the 2021 SIAM International Conference on Data Mining (SDM). Society for Industrial and Applied Mathematics, 2021: 333-341.
>
> [32] Pan Li, Yanbang Wang, Hongwei Wang, and Jure Leskovec. Distance encoding: Design provably more powerful neural networks for graph representation learning. In Advances in Neural Information Processing Systems, volume 33, 2020.
>
> [S3] Morris C, Rattan G, Mutzel P. Weisfeiler and Leman go sparse: Towards scalable higher-order graph embeddings[J]. arXiv preprint arXiv:1904.01543, 2019.
>
> [S4] Morris C, Ritzert M, Fey M, et al. Weisfeiler and leman go neural: Higher-order graph neural networks[C]//Proceedings of the AAAI Conference on Artificial Intelligence. 2019, 33(01): 4602-4609.
>
> [S5] H. Maron, H. Ben-Hamu, H. Serviansky, et al. Provably powerful graph networks, in Advances in Neural Information Processing Systems, 2019, pp. 2153–2164.

---

> ### Comment · Reviewer_ULms · 2021-08-19
> **follow up question**
>
> I thank the reviewers for their detailed response. I am still somewhat confused here w.r.t. Q1 and Q2.
>
> Specifically, the WL algorithm is employed as a main subroutine by "nauty" (and most other solvers of that kind see e.g., the discussion here https://arxiv.org/pdf/2005.10182.pdf), and nauty is in fact an isomorphism solver. It is clear to me that neighborhood isomorphism is not the same as an automorphism, in general, but if I let the neighborhood grow large enough I get an efficient automorphism heuristic?!
> Could the authors elaborate on this?

---

> > ### Author Response · Authors · 2021-08-20
> > **Response to Follow-up Question**
> >
> > Thanks for the feedback. You are right that 1-WL is an important subroutine of Nauty algorithm, but 1-WL itself is not sufficient to detect Automorphic Equivalence (AE). We will explain from the following two points:
> >
> > **A) Nauty algorithm goes beyond WL test.** 1-WL, also known as color refinement or partition refinement, is only part of the solution in Nauty algorithm [1, 2]. One good example is provided in (https://pallini.di.uniroma1.it/Introduction.html#!prettyPhoto). Please check out the figures in block 5. WL test alone will be trapped in the third figure (EQUITABLE PARTITION) in block 5. Because nodes {3, 4, 5, 6} all connect to 2 dark green nodes and 1 orange node, WL algorithm cannot differentiate them. However, {3, 4} and {5, 6} belong to different AE sets as the fourth figure (ORBIT PARTITION) showed. An additional "individualization-refinement" method is used to further detect automorphisms, which is shown in 5.1 and 5.2. It respectively sets the tested node pair (*e.g.*, {3, 4}) to a unique color and then iteratively updates the colors of the rest of nodes. It determines two nodes are AE if the derived colored graphs are isomorphic, which effectively differentiates {3, 4} and {5, 6}. This "individualization-refinement" method goes beyond WL algorithm.
> >
> > **B) AE is a stricter condition than neighborhood isomorphism for arbitrary neighborhood size.** This was proven as the Theorem 1 in [3]. An example of regular graph with girth 12 and average node degree 3, *i.e.*, (3,12) cage, was constructed in [4]. There are two different AE sets in this example graph, but nodes from each AE set are neighborhood isomorphic even if they consider the entire graph as neighborhood. The reason is "the isomorphism needs not to map one ego onto the other".
> >
> > Now, we can look at the full picture here. Previous work showed MPNN can be at most as powerful as WL test [5], while WL test can only solve part of the isomorphism problems [6]. Besides, isomorphism is a less condition compared to AE. That is, **MPNN <= WL test < Isomorphism < AE**. Since our model is provably expressive in capturing AE feature, it will be strictly more powerful than the GNN variants that follow the hierarchy of WL test.
> >
> > We apologize for the unclear presentation. We will improve it in the final version. And, we are happy to discuss more if you still have doubts.
> >
> > [1] Kiefer S, McKay B D. The iteration number of colour refinement[J]. arXiv preprint arXiv:2005.10182, 2020.
> >
> > [2] McKay B D, Piperno A. Practical graph isomorphism, II[J]. Journal of symbolic computation, 2014, 60: 94-112.
> >
> > [3] Martin G Everett, John P Boyd, and Stephen P Borgatti. Ego-centered and local roles: A graph theoretic approach. Journal of Mathematical Sociology, 15(3-4):163–172, 1990.
> > [4] Biggs N, Biggs N L, Norman B. Algebraic graph theory[M]. Cambridge university press, 1993.
> >
> > [5] Xu K, Hu W, Leskovec J, et al. How Powerful are Graph Neural Networks?[C]//International Conference on Learning Representations. 2018.
> >
> > [6] Kiefer S, Immerman N, Schweitzer P, et al. Power and limits of the Weisfeiler-Leman algorithm[R]. Fachgruppe Informatik, 2020.

---

> > > ### Comment · Reviewer_ULms · 2021-08-23
> > > **..**
> > >
> > > Thanks for your additional comments.
> > > I agree with those points, however, since the nauty algorithm is applied here, it is not clear how much stronger the architecture is in practise, i.e., we have more representational power of WL but it is not clear how much more -- at least from a theory perspective. From a practical perspective the experiments are quite promising.

---

### Author Response · Authors · 2021-08-10
**Response to Reviewer ULms (Part II)**

**Q5.** *the paper would benefit from the inclusion of a larger array of datasets*

**Response:** Thanks for the suggestion. Our proposed model aims to capture the structural equivalence in graph analysis, which plays important roles in social network, scientific community network and behavior network. Therefore, we choose 8 real-world networks collected from 3 domains, *i.e.*, Facebook friendship, scientific citation and Amazon purchase, to evaluate our model. Besides, we find classic and recent GNN research evaluate their model on similar scale of datasets (see the table below). Therefore, we feel that our experiments are sufficiently representative. Following your suggestion, we will try to add in more benchmark datasets in the final version and also after the publication.

```
GraphSage (NeurIPS 2017):
    # datasets : 3;
    # domains  : 3;
GIN (ICLR 2019):
    # datasets : 9;
    # domains  : 2;
De-GNN (NeurIPS 2020):
    # datasets : 6;
    # domains  : 2;
PNA (NeurIPS 2020):
    # datasets : 3;
    # domains  : 2;
LRP (NeurIPS 2020):
    # datasets : 3;
    # domains  : 1;
GRAPE (ours):
    # datasets : 8;
    # domains  : 3;
```

**Q6.** *Given the broad literature on limitations of GNN (in terms of WL test, robustness guarantees, etc), I find it surprising that a discussion of the weaknesses of the presented method is not really present. The possible societal impact of using graph neural networks in classification task (e.g., in social networks, as considered here) is also not discussed in any detail.*

**Response:** Thanks for the feedback. One potential limitation of our model is its scalability to large-scale real-world applications (see complexity analysis in Appendix C.3), which is a common challenge to many GNN models. However, since our model is defined on localized automorphic equivalence, sampling technique similar to GraphSage can be designed to ensure the feasibility of computation overhead. We will leave this as a future work. Besides, our paper proposes a general GNN framework and it is evaluated on public available datasets. Therefore, we feel that the negative societal impact is manageable. We will add in discussion and software license to advocate appropriate usage. Following your suggestion, we will add in paragraphs to elaborate on the limitation and societal impact of our work.

---

### Author Response · Authors · 2021-08-10
**Response to Reviewer nRJR**

**Q1.** *The method using automorphism equivalence matching and counting to improve the expressive power of GNNs is previously proposed in [6], while the relation of this work to [6] is not adequately addressed and discussed. [6] also proved by matching and counting subgraph isomorphism, the expressive power of GNNs can be improved. Though the AE-aware aggregator is not the same as the counting based aggregation in [6], the similarity in the core idea should definitely be discussed more.*

**Response:**

First, please kindly note that the ref [6] mentioned by the reviewer is put on arxiv but not published on peer-reviewed platform yet. A more detailed comparison is as follow.

1. As we discussed in Related Works section, the GNN variants beyond MPNN framework can be mainly classified as augmenting node features and designing more powerful architecture. The related work [6] belongs to the first category, since it proposes to add substructure counts into node’s feature. On the other hand, our model falls into the category of designing more powerful architecture. More importantly, as we show in Table 1 and Table 4 (in appendix), our model is orthogonal to node feature since it achieves the best performance on the datasets with both original and dummy node feature. Therefore, the related work [6] is not a competitor to our model, but we can also strengthen our model with the augmented node feature.

2. The authors in related work [6] claimed their method preserve the locality of message passing, because the augmented node feature can be feed into standard MPNN aggregator. However, the augmented node feature itself is global invariant. That is, given a set of subgraph templates, the nodes will pass the same feature to all their neighbors, which is the count of their membership in these templates. On the contrary, we propose a localized version of automorphic equivalence (Ego-AE) and principally integrate it into the GNN framework. It allows GNN to differentiate the **relative** structure roles in each node’s local neighborhood, which is essential to learn expressive node representations. Take Figure 2 for example, the subgraph counting method with triangle template will differentiate v2, v5, v6 (counting 2 triangles) from v3, v4 (counting 1 triangle). However, when learning the representation for node v1, our Grape model can capture their similar roles in v1's local neighborhood as they all form triangle structure with v1. Therefore, the substructure counting method cannot differentiate the Ego-AE feature we proposed, which is essential for GNN to differentiate the semantic-aware influences coming from each node's different neighbors.

3. The local nature of our model also allows us to propose efficient incremental subgraph matching and genetic subgraph template search algorithms. These algorithms ensure our model is independent of hand-craft templates and can be applied to large-scale networks. When applying to large-scale real-world problems, our model can also adopt sampling technique similar to GraphSage to limit the number of matched instance starting from each node, which can ensure the computation feasibility of our model. On the contrary, the simple counting method in [6] does not allow for efficient matching or automatic template search.

To conclude, we believe ref [6] does not weaken our work's novelty in proposing an efficient and powerful GNN architecture that captures automorphic equivalences. Thanks for the constructive comments. Following your suggestion, we will elaborate on our model's connection to [6] in the final version.

>[6] Giorgos Bouritsas, Fabrizio Frasca, Stefanos Zafeiriou, and Michael M Bronstein. Improving graph neural network expressivity via subgraph isomorphism counting. arXiv preprint arXiv:2006.09252, 2020.

---

> ### Comment · Reviewer_nRJR · 2021-08-31
> **Response**
>
> Thanks for the authors' response. I agree that the aggregation method in this work is novel and this work contributes a subgraph template search method which [6] didn't have. I'll raise my rating to 6. But heavy discussion about [6] should be definitely included in the later version of this paper.

---

### Decision · Program_Chairs · 2021-09-27

**Decision:**

Accept (Poster)

**Comment:**

This paper presents a new GNN architecture which addresses the automorphic equivalence among nodes. The idea is novel and effective. The results have demonstrated the superiority of the proposed approach. The reviewers liked the idea, while raised concerns such as some theoretical claims, datasets and the selection of subgraph templates. The authors have done a great job in rebuttal and have clarified some claims that caused misunderstanding. In the end, the reviewers reached consensus in accepting this paper.